Journal of Data-centric Machine Learning Research (2024)          Submitted 11/23; Revised 11/24; Published 12/24

# V-LoL😂: A Diagnostic Dataset for
# Visual Logical Learning

**Lukas Helff**[1,2] *                                LUKAS.HELFF@CS.TU-DARMSTADT.DE

**Wolfgang Stammer**[1,2] *                         WOLFGANG.STAMMER@CS.TU-DARMSTADT.DE

**Hikaru Shindo**[1]                                HIKARU.SHINDO@CS.TU-DARMSTADT.DE

**Devendra Singh Dhami**[2,3]                        D.S.DHAMI@TUE.NL

**Kristian Kersting**[1,2,4,5]                       KRISTIAN.KERSTING@CS.TU-DARMSTADT.DE

[1]AI AND ML GROUP, TU DARMSTADT; [2]HESSIAN CENTER FOR AI (HESSIAN.AI)
[3]EINDHOVEN UNIVERSITY OF TECHNOLOGY;
[4]CENTRE FOR COGNITIVE SCIENCE, TU DARMSTADT; [5]GERMAN CENTER FOR AI (DFKI)

**Reviewed on OpenReview:** HTTPS://OPENREVIEW.NET/FORUM?ID=IKBFIPIQFE

**Editor:** Christopher De Sa

## Abstract

Despite the successes of recent developments in visual AI, different shortcomings still exist; from missing exact logical reasoning, to abstract generalization abilities, to understanding complex and noisy scenes. Unfortunately, existing benchmarks, were not designed to capture more than a few of these aspects. Whereas deep learning datasets focus on visually complex data but simple visual reasoning tasks, inductive logic datasets involve complex logical learning tasks, however, lack the visual component. To address this, we propose the diagnostic visual logical learning dataset, V-LoL, that seamlessly combines visual and logical challenges. Notably, we introduce the first instantiation of V-LoL, V-LoL🚂, – a visual rendition of a classic benchmark in symbolic AI, the Michalski train problem. By incorporating intricate visual scenes and flexible logical reasoning tasks within a versatile framework, V-LoL🚂 provides a platform for investigating a wide range of visual logical learning challenges. We evaluate a variety of AI systems including traditional symbolic AI, neural AI, as well as neuro-symbolic AI. Our evaluations demonstrate that even SOTA AI faces difficulties in dealing with visual logical learning challenges, highlighting unique advantages and limitations of each methodology. Overall, V-LoL opens up new avenues for understanding and enhancing current abilities in visual logical learning for AI systems. All code and data is available at HTTPS://SITES.GOOGLE.COM/VIEW/V-LOL.

---

*. equal contribution

## 1 Introduction

As humans, we effortlessly integrate visual perception with logical reasoning to make sense of the world around us, answering questions both in daily and complex situations. Whether it's astronomers deciphering the cosmos, medical scientists analyzing scans, or car drivers navigating the roads, individuals synthesize logical concepts with visual observations to make informed decisions and execute actions. Achieving such a seamless integration between vision and logical reasoning remains a longstanding goal in the realm of visual AI. Several works attempt to tackle this issue by separating the perceptual and logical processing from another *e.g.*, via neural image encoders that perform accurate multi-label prediction followed by (exact) logical inference methods (Shindo et al., 2023; Mao et al., 2019; Vedantam et al., 2019). Other approaches, on the other hand, focus on joint representations from the start *e.g.*, via large multi-modal models in which perception and logical reasoning is intertwined within one large model (Rose et al., 2023; Chen et al., 2023; Zhang et al., 2023). Which of these directions will prevail in the long run is still very much an open question. Regardless of the direction, however, the process of developing AI models that can handle visual logical learning and the multitude of its subproblems requires extensive diagnostic tests to analyze progress and discover individual shortcomings.

While related deep learning (DL) datasets predominantly address visual perception challenges (Lin et al., 2014; Cordts et al., 2016; Karazija et al., 2021; Schuhmann et al., 2022), there is a burgeoning interest in higher-level reasoning tasks (Antol et al., 2015; Johnson et al., 2017; Hong et al., 2021; Zellers et al., 2019; Hudson and Manning, 2019) such as scene understanding or visual question answering (VQA). Yet, most of these require only simple reasoning abilities. On the other hand, traditional inductive logic programming benchmarks (Michalski, 1980; Michie et al., 1994; Dua and Graff, 2017) aim to tackle more complex reasoning abilities but neglect the visual component that is required for AI models to interact in our complex visual surroundings. More recent datasets have begun introducing more sophisticated reasoning problems into visual datasets, *e.g.*, logical concept learning (Vedantam et al., 2021) and analogical visual reasoning (Zhang et al., 2019). While this development marks a positive direction, these *benchmark* datasets, however, often consist of fixed sample sets and are designed to cover only limited areas within the domain of visual logical learning. Moreover, they lack a versatile framework that allows researchers to design and generate custom-tailored problem sets to diagnose models over a broad range of challenges.

In this paper, we therefore introduce the Visual Logical Learning diagnostic dataset (V-LoL). Due to its flexibility and easy, modular generation V-LoL specifically allows to study the ability of visual AI models in a wide-range and individual visual-logical learning aspects. The fundamental idea of V-LoL remains to integrate the explicit logical learning tasks of classic symbolic AI benchmarks into visually complex scenes, creating a unique visual input that retains the challenges and versatility of explicit logic. By doing so, V-LoL bridges the gap between symbolic AI challenges and contemporary deep learning datasets offering various visual logical learning tasks, which allow for extensive evaluations of AI models across a wide spectrum of AI research.

We specifically introduce the V-LoL⊞ dataset, which comprises two distinct instantiations of V-LoL: V-LoL-Trains (V-LoL⊞) and V-LoL-Blocks (V-LoL□). Both are conceived as image classification datasets using the symbolic representations of the Michalski train

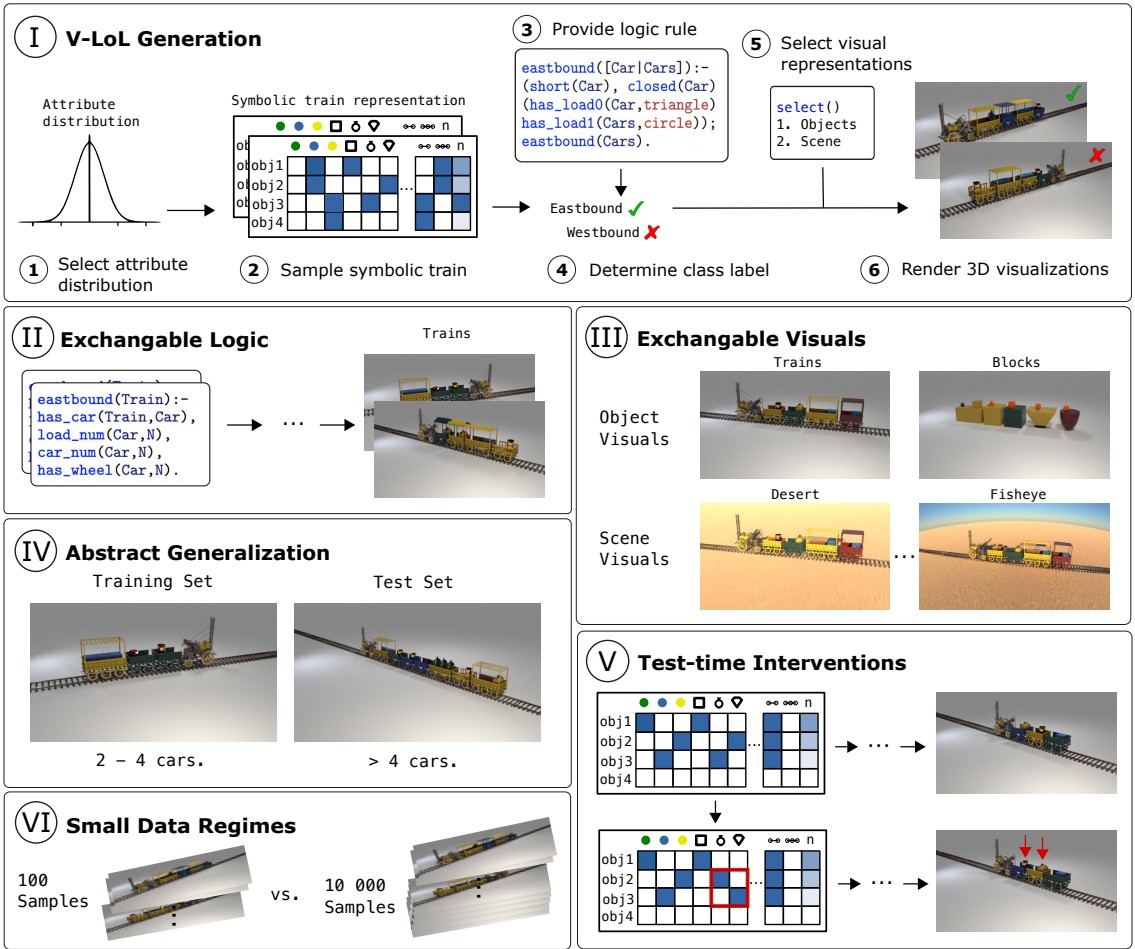

Figure 1: V-LoL: a diagnostic dataset that allows to test a variety of visual logical learning challenges. (I) The modular generation process of V-LoL consists of sampling symbolic train representations (*i.e.*, train cars and their attributes) from a pre-defined distribution. Via a logical class rule the class affiliation of each train sample is determined. The 3D visual representations are selected and finally rendered. The flexibility and versatility of V-LoL allows that the logic component (II) as well as the visual component (III) can easily be exchanged. This way one can perform different visual-logical learning tests *e.g.*, concerning abstract generalization abilities (IV), targeted test-time interventions (V) or evaluations on varying dataset sizes (VI).

problem (Michalski, 1980) which are rendered in a CLEVR-like fashion (Johnson et al., 2017). Hereby, in contrast to the relatively straightforward visual reasoning task offered by CLEVR, V-LoL leverages the logical foundation of the Michalski train problem to establish more intricate visual logical learning problems. The V-LoL generator seamlessly integrates any provided logic rule into appealing, yet complex images (*cf.* Fig. 1 (I)), enabling precise control over the visual and logical task difficulty. The generated tasks can encompass a wide range of challenges, including object recognition, counting, interpretation

of spatial arrangements, comprehension of arithmetic and logical operators, and identifying and decoding intricate, chained reasoning patterns (*cf.* Fig. 1). In our evaluations we show V-LoL's flexibility and versatility in generating several of such challenges and are able to reveal benefits and shortcomings arising from different symbolic, neural and neuro-symbolic AI approaches.

Overall, we make several important contributions: (i) We introduce the Visual Logical Learning diagnostic dataset (V-LoL) that brings logic-based ILP benchmarks into the realm of deep learning. (ii) We present V-LoL, an initial instantiation of V-LoL that builds upon the logical foundation of the Michalski train problem and the immersive environment of CLEVR to establish a modern AI dataset that is relevant across the spectrum of AI research. (iii) The provided V-LoL generator offers great flexibility within the visual and logical components, allowing to seamlessly integrate any arbitrary logic program into a new visually complex dataset. (iv) We provide a flexible and user-friendly framework, allowing for a straightforward generation of large-scale visual datasets with rich scene information. (v) By evaluating various symbolic, neural, and neuro-symbolic AI models on several possible V-LoL challenges, we offer insights into their visual logical learning abilities, exemplifying the value of V-LoL for investigating shortcomings and thereby helping improve AI models.

## 2 V-LoL: Merging Logic and Vision

Overall, the V-LoL diagnostic dataset unites the challenges of logical reasoning with the complexity of visual perception. In spirit of the never-ending language learning framework (NELL) proposed by Carlson et al. (2010), V-LoL serves as an overarching framework for datasets that lift classical logic learning problems (*e.g.*, Inductive Logic Programming (ILP)) into a detailed visual environment. In an ILP problem a logic rule is learned given a set of positive and negative examples as well as background knowledge, where the examples and knowledge are represented as logical formulae (Muggleton, 1991; Cropper and Dumančić, 2022). V-LoL's unique datasets facilitate the evaluation and development of AI models, particularly focusing on their ability to perform logical inferences within a visual environment. Thus, by merging logic and vision, V-LoL allows to simultaneously evaluate and diagnose both the visual perception and logical reasoning abilities of AI systems.

V-LoL-Trains (V-LoL) presents a first instantiation of V-LoL by merging the logical foundation of the Michalski train problems (Michalski, 1980) into 3D visual representations via CLEVR-like (Johnson et al., 2017) rendering processes. Introduced by Michalski (1980), the original *Michalski train problem* represents a classic ILP task. It consists of two sets of five train examples where these trains are composed of a wide variety of properties, and are labelled into two categories: *eastbound* or *westbound* (*cf.* Fig. 5 in the supplementary material (supp.)). The objective is to discern relational patterns within the trains' properties and conjecture a hypothesis that perfectly separates the eastbound from the westbound trains. We will provide details on the exact composition and properties of V-LoL in the following sections.

### 2.1 V-LoL Generation

Generating a V-LoL image is performed in a six-step generation process. This process is outlined in pseudo-code in Alg. 1 and is aligned with the steps sketched in Fig. 1 (top).

---

**Algorithm 1** V-LoL train generation for a single image.

1. distr ← SELECT_DIST() // *User selects attribute distribution*
2. symb_train ← SAMPLE_SYMB_TRAIN(distr) // *Sample symbolic train representations*
3. logic_program ← DEFINE_LOGIC() // *User defines rule as logic program*
4. class ← EVAL(symb_train, logic_program) // *Evaluate class affinity based on logic program*
5. visuals ← SELECT_VISUALS() // *User selects background and object visuals*
6. v_lol_train ← GENERATE(symb_train, class, visuals) // *Create scene and render image*

---

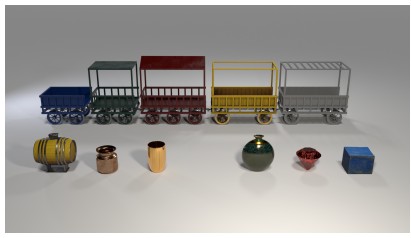

V-LoL⬚

| Car Position | Car Colour | Car Length | Car Wall | Car Roof | Car Axles | Load Num. | Load Shape |
|---|---|---|---|---|---|---|---|
| 1 | Yellow | Short | Full | None | 2 | 0 | Blue Box |
| 2 | Green | Long | Railing | Frame | 3 | 1 | Golden Vase |
| 3 | Grey | | | Flat | | 2 | Barrel |
| 4 | Red | | | Bars | | 3 | Diamond |
| | Blue | | | Peaked | | | Metal Pot |
| | | | | | | | Oval Vase |
| | | | | | | | None |

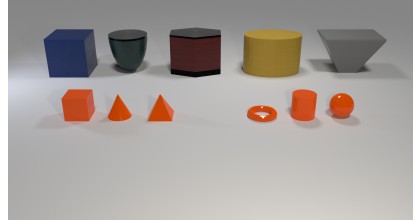

V-LoL□

| Car Position | Car Colour | Car Length | Black Top | Car Shape | Black Bottom | Load Num. | Load Shape |
|---|---|---|---|---|---|---|---|
| 1 | Yellow | Short | True | Cube | True | 0 | Sphere |
| 2 | Green | Long | False | Cylinder | False | 1 | Pyramid |
| 3 | Grey | | | Hemisphere | | 2 | Cube |
| 4 | Red | | | Frustum | | 3 | Cylinder |
| | Blue | | | Hex. Prism | | | Cone |
| | | | | | | | Torus |
| | | | | | | | None |

Figure 2: Right: a detailed overview of the train (top) and block (bottom) representations. Left: illustrations of the visual representations of individual objects and attributes used for rendering the symbolic representations into images.

Briefly, one first selects a distribution for the train attributes including the length of the train. Second, a valid symbolic representation of a train composition (symbolic train) is randomly sampled from this distribution. Third, one defines an underlying logic rule which the train within the image should depict. This logic rule, represented as a logic program, is based on the composition and relation of the train's parts. Fourth, via the the logic program the class label is derived for the sampled symbolic train. Fifth, one selects the desired visual representations of the image, *e.g.*, the background scene and the object visualizations. Finally, the 3-D V-LoL⬚ image is generated based on the sampled symbolic train and selected visual representations. We will discuss these steps in detail in the following.

**Symbolic trains** (steps 1 and 2 of Alg. 1). The symbolic train of the V-LoL⬚ generation process is based on objects and relationships that are semantically similar to those of the prolog representation of Muggleton (Muggleton, 1998) (*cf.* Tab. 2 of supp.). Accordingly, while the trains have different attributes and associated allocations, the overall composition, number of attributes, and number of corresponding allocations (groundings), remain identical (*cf.* Fig. 2 (top) for more detailed information on the individual attributes). To generate symbolic trains, V-LoL⬚ allows sampling from two different attribute distributions (*cf.* step 1 of Alg. 1), namely the *Michalski train* and the *random train* (*i.e.*, uniform)

distribution. With the *Michalski train* distribution, V-LoL aims to preserve statistical coherence to the original Michalski trains (Michalski, 1980). To do so, we integrate the prolog train-generator from Muggleton (1998), which incorporates assumptions regarding the value distribution of the train attributes and follows the constraints outlined in the supplement (*cf.* Sec. A.2). In contrast, the *random trains* distribution does not impose any assumptions regarding the distribution of train attributes and only apply attribute constraints necessary for visualization (*e.g.*, a short car cannot accommodate more than two payloads). Independent of the distribution, the user can specify a fixed length or a length interval from which the train length is randomly selected. Upon selection of the desired attribute distribution in step 1, a symbolic train is randomly sampled from the chosen distribution in step 2 of Alg. 1.

**Programmable logic** (steps 3 and 4 of Alg. 1). Programmable logic, in this work, refers to a framework that allows to define and modify logical rules, *and* to perform computations and operations based on these. It is an integral component of V-LoL that enables the implementation of logic programs (*cf.* step 3 of Alg. 1) and their integration into the dataset generation process (*cf.* step 4 of Alg. 1). In step 3 of Alg. 1, users are provided with a predefined vocabulary of Prolog predicates for constructing individual logic programs (users further have the option to expand this vocabulary via self-defined predicates). Based on these predicates users define a logic rule (which the trains of their dataset should depict) in the form of a logic program. V-LoL next evaluates the class affinity for the previously sampled symbolic train (*cf.* step 4 of Alg. 1) given this logic program. Hereby, a train sample is considered to be "eastbound" if the configuration of its parts is in accordance with the defined logic rule. If this is not so, the train is considered to be "westbound" (*cf.* Fig. 1 (top)).

Overall, by defining various logic rules, users can create a wide range of distinct problems to evaluate AI models regarding their capabilities in visual logical learning. For our experimental evaluations, we have designed a diverse set of logical challenges to analyze and evaluate the limitations of current AI models. The logic rules employed in these challenges are defined below. The corresponding Prolog and first-order logic (FOL) formulations, along with further details regarding the underlying properties of each rule, can be found in Section B.3 and Table 3 (of the supp.), respectively.

(i) **Theory X**: The train has either a short, closed car or a car with a barrel load is somewhere behind a car with a golden vase load. This rule was originally introduced as "Theory X" in the new East-West Challenge (Bloedorn et al., 1995). We provide the First Order Logic (FOL) and Prolog definitions below:

FOL:

$$eastbound(Train) \models \exists Car_1, Car_2,$$
$$has\text{-}car(Train, Car_1) \land has\text{-}car(Train, Car_2)$$
$$\land ((short(Car_1) \land closed(Car_1))$$
$$\lor (has\text{-}load(Car_1, golden\text{-}vase)$$
$$\land has\text{-}load(Car_2, barrel)$$
$$\land somewhere\text{-}behind(Train, Car_2, Car_1)))$$

Prolog:

```
eastbound([Car|Cars]):-
(short(Car), closed(Car));
(has_load0(Car,golden-vase),
has_load1(Cars,barrel));
eastbound(Cars).
```

Varying Background Scenes · Rich Scene Information

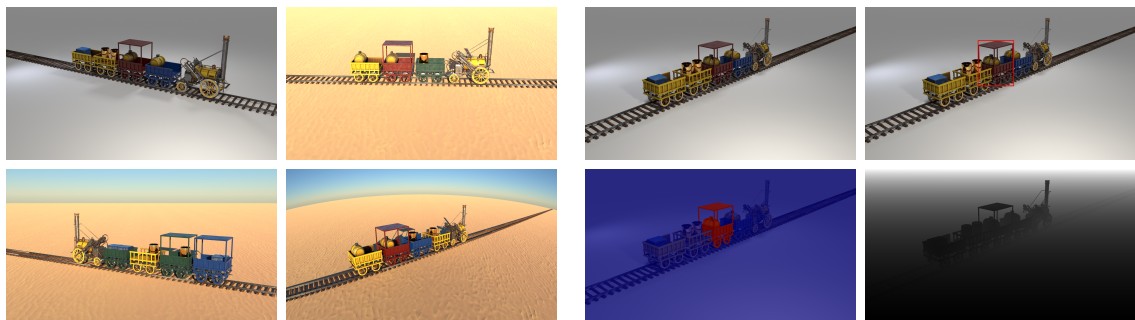

Figure 3: Different background scenes and rich scene information provided with each V-LoL⊡ sample. The four background options consist of (left): a base, desert, desert with sky, and fisheye scene. The scene information (right) provided with each sample contains: the original image sample, object bounding boxes, pixel-level object segmentation maps and a corresponding depth map.

(ii) **Numerical rule**: The train has a car where its car position equals its number of payloads which equals its number of wheel axles.

(iii) **Complex rule**: Either, there is a car with a car number which is smaller than its number of wheel axles count and smaller than the number of loads, or there is a short and a long car with the same colour where the position number of the short car is smaller than the number of wheel axles of the long car, or the train has three differently coloured cars.

**Generating images from symbolic trains** (steps 5 and 6 of Alg. 1). A main component for bringing classic, symbolic AI benchmarks into the realm of deep learning is to convert the initially purely symbolic representations into complex visual scenes. To do so V-LoL⊡ proceeds with the following two steps. In step 5 of Alg. 1, users have the option to select from different visual representations. Specifically, V-LoL⊡ offers two distinct types. On one hand, V-LoL⊡, which comprises images depicting detailed trains that are reminiscent of model trains. On the other hand, V-LoL-Blocks (V-LoL□), draws inspiration from the minimalist aesthetics of the CLEVR dataset (Johnson et al., 2017) and presents a simpler visual style resembling children's building blocks. Fig. 2 provides an overview of both options, displaying their respective attributes in tables (right) and illustrating these through images (left). Notably, the number of objects, attributes, and logical versatility remains identical between both, what varies is the complexity of the visual representations. Additionally, the generation process offers four background scenes that are compatible with both the trains and blocks representations (*cf.* step 5 of Alg. 1). These scenes include a basic scene, a desert scene, a desert scene with sky, and a fisheye background, each featuring different levels of texture, distortions, and noise. We refer to Fig. 1 and 3 (left) for their visualizations.

In step 6 of Alg. 1, the sampled symbolic trains are rendered into a CLEVR-based (Johnson et al., 2017) environment. During this process, individual trains are constructed according to the previously selected object representations. Subsequently, they are placed in the foreground of the chosen background scene and rendered using the Blender engine (*cf.* Sec. E

Table 1: A comparison between V-LoL and other visual reasoning datasets. We here differentiate between: 3D visualization, compositional objects (objects that consist of other objects), rich scene information (*e.g.*, depth maps, pixel-wise segmentations, etc.), variable number of test objects (in comparison to training set), inclusion of programmable logic (PL), reasoning about relations, arithmetics, and analogies, and ILP problem.

| Dataset | 3D | Comp. Objs. | Rich Scene Information | Variable # Test Objs. | Diagnostic | PL | Relation | Arithmetic | Analogy | ILP |
|---|---|---|---|---|---|---|---|---|---|---|
| VQA | ✓ | ✓ | ✗ | ✗ | ✗ | ✗ | ✓ | ✗ | ✗ | ✗ |
| CLEVR | ✓ | ✗ | ✗ | (✓) | ✓ | ✗ | ✓ | ✗ | ✗ | ✗ |
| CLEVR-Hans3 | ✓ | ✗ | ✗ | (✓) | ✗ | ✗ | ✓ | ✗ | ✗ | ✗ |
| CURI | ✓ | ✗ | ✗ | (✓) | ✗ | ✗ | ✓ | ✗ | ✓ | ✗ |
| ACRE | ✓ | ✗ | ✓ | (✓) | ✓ | ✗ | ✓ | ✗ | ✓ | ✗ |
| PTR | ✓ | ✓ | ✓ | ✗ | ✓ | ✗ | ✓ | ✓ | ✓ | ✗ |
| Bongard-LOGO | ✗ | ✗ | ✗ | ✗ | ✗ | ✗ | ✓ | ✗ | ✓ | ✗ |
| Kandinsky | ✗ | ✗ | ✗ | (✓) | ✗ | ✗ | ✓ | ✗ | ✓ | ✗ |
| RAVEN | ✗ | ✓ | ✗ | ✗ | ✓ | ✗ | ✓ | ✓ | ✓ | ✗ |
| **V-LoL** | ✓ | ✓ | ✓ | ✓ | ✓ | ✓ | ✓ | ✓ | ✓ | ✓ |

in the supplements for details). This results in a `v_lol_train` sample which consists of the rendered image and its corresponding class label (*cf.* Fig. 1 (top)). Furthermore, aach image of V-LoL comes with rich ground-truth object information. Specifically, for each train car attribute, we provide comprehensive information including: the position within the scene determined by the centre of the three dimensional object, a binary pixel-wise mask emphasizing the pixels correlated with the train attribute, and its two-dimensional bounding box that encloses the mask. The depth information for each image are additionally stored. Fig. 3 (right) depicts annotations for an example image.

**Generating a set of images.** Finally, to generate a set of V-LoL images that contain an equal ratio of east- and westbound trains, V-LoL employs rejection sampling over steps 2 and 4. *I.e.*, we sample symbolic trains from the selected distribution in step 2 based on the class affinities as identified in step 4 until we have reached the desired balance between east- and westbound samples. This additional loop over step 2 and 4 is employed before the images are generated.

## 2.2 V-LoL and Related Datasets

V-LoL not only combines high visual complexity with intricate logical challenges but also introduces a flexible framework that offers precise control over both components. Specifically, unlike other datasets that are often constrained to fixed sets of challenges, our framework utilizes varying visual representations with programmable logic (PL) that allows experimentalists to design and generate custom sets of visual logical learning problems. These can thus be tailored to the experimentalist's specific requirements and thereby facilitate extensive and targeted diagnostic evaluations, *e.g.*, on specific aspects of relational, arithmetic or analogical reasoning. Additional noteworthy properties of V-LoL are that the images comprise hierarchical objects (objects composed of "sub-objects") and that the flexibility of

the framework allows for a user to specify different training and test settings, *e.g.*, the test set can contain a greatly different amount of objects than occur within the training set.

Over the full set of properties, V-LoL⟑ thus offers several advantages over previous related datasets. Accordingly (*cf.* Tab. 1), classic visual question answering datasets, whether based on natural images (VQA (Antol et al., 2015)) or synthetic images (CLEVR (Johnson et al., 2017)), may focus on complex visual representations, but focus on a more narrow set of logical problems than V-LoL⟑. The same holds for CLEVR-Hans (Stammer et al., 2021), a confounded classification dataset that is based on CLEVR visuals. While the aforementioned datasets primarily focus on straightforward reasoning tasks, others like CURI (Vedantam et al., 2021), ACRE (Zhang et al., 2021), and PTR (Hong et al., 2021), introduce more sophisticated reasoning tasks involving analogical or even arithmetical reasoning. These datasets follow the CLEVR-like approach in image generation, featuring simple geometric structures that, however, feature less visually complex scenes.

Additionally, the images of these datasets encompass a much smaller combinatorial complexity than V-LoL⟑ images. *E.g.*, ACRE consists of 48 object permutations, where a V-LoL⟑ image carrying just one car already encompasses 22000 combinations. This further grows exponentially with increasing train lengths. On another end of the spectrum, Bongard-LOGO (Nie et al., 2020), Kandinsky patterns (Holzinger et al., 2019, 2021), and RAVEN (Zhang et al., 2019) are even more centered on complex reasoning challenges, yet largly neglect visual complexity, *i.e.*, featuring only elementary two-dimensional objects. Lastly and arguably more importantly, unlike the majority of previously mentioned, datasets that focus only on a fixed subset of visual logical learning challenges, V-LoL represents a valuable diagnostic tool for probing the full depths of visual and logical learning in a controlled, yet versatile environment.

## 3 Experiments: AI Systems on the V-LoL challenges

In our experimental evaluations we showcase V-LoL's versatility for providing diagnostic tests for visual logical learning abilities. Specifically, we generate datasets via V-LoL⟑ for different tasks within visual logical learning and evaluate several AI approaches on these. Before we dive into the specific evaluations on these *challenges*, we first give an overview on the experimental setup as well as investigated models.

**AI Models.** We evaluate AI models from neural, symbolic, and neuro-symbolic AI where we have deliberately chosen methods that encompass a wide range of AI approaches, aiming to gain comprehensive insights into their capabilities and to explore the advantages and disadvantages arising from their different methodologies.

AI approaches that fall under the term **Neural AI** perform inference on an implicit, subsymbolic-level knowledge representation and have become the prevailing paradigm in recent years, particularly in visual AI. However, despite their promising predictive performances these approaches have also been shown to be strongly influenced *e.g.*, by shortcut behaviour (Schramowski et al., 2020) and biases (Bender et al., 2021), and the degree to which they perform logical reasoning is still an open research topic (Wei et al., 2022; Creswell et al., 2022; Saparov and He, 2022). In our evaluations we specifically assess the abilities of *ResNet18* (He et al., 2016), *EfficientNet* (Tan and Le, 2021), and Vision

Transformer (Dosovitskiy et al., 2021) (*ViT*) in handling visual logical learning challenges. We perform additional experiments on large language models (LLMs) including Llama2 (Touvron et al., 2023) and ChatGPT (OpenAI, 2023).

**Symbolic AI** approaches (aka Good Old-Fashioned AI (GOFAI) (Russell and Norvig, 2020)) perform inference on explicit, high-level symbol-based knowledge representations making them well-suited for tasks that require logical reasoning and rule-based decision-making, but ill-suited for inference on low-level data such as raw images. We specifically investigate the abilities of the classical ILP approach, Aleph (Srinivasan, 2001) (*Aleph (GT)*), and more recent approach, Popper (Cropper and Morel, 2020) (*Popper (GT)*) by evaluating their induced programs on the ground-truth symbolic representations of the dataset. These evaluations act as a form of ablation given the presence of an omniscient perception model.

**Neuro-Symbolic AI** encompasses a wide range of approaches that follow the idea of combining neural (subsymbolic) with symbolic computations. The motivation behind this is to combine the strengths of both approaches and as such mitigate the shortcomings of the individual methods. The neuro-symbolic models in our evaluations can be categorized into two subcategories (following the categorization of (Kautz, 2022)): Neuro|Symbolic (*RCNN-Popper* and *RCNN-Aleph*) and Neuro:Symbolic→Neuro ($\alpha ILP$ (Shindo et al., 2023)). Where $\alpha$ILP utilizes gradients to learn logic rules, *i.e.*, differentiable ILP framework, RCNN-Aleph and RCNN-Popper combine their base ILP approach with a Mask-RCNN model (He et al., 2017) that is pretrained on randomly distributed train images for object identification and attribute prediction. In this way, the Mask-RCNN model infers the symbolic representation from an image, that serves as the symbolic input for both ILP approaches. Thus, in comparison to Aleph (GT) and Popper (GT), RCNN-Aleph and RCNN-Popper can contain possible perception errors stemming from the Mask-RCNN module.

**Experimental setup.** Unless stated otherwise the problem setup in our evaluations is a classification setup consisting of a training dataset and held-out test set of images belonging to one of two classes (eastbound and westbound). Furthermore, unless stated otherwise the training set includes 1k images. All models are trained on specified training splits and evaluated by performing stratified 5-fold cross-validation on a held-out test set, containing 2k images. Quantitative results (unless noted otherwise) correspond to the test set classification accuracy. The hyperparameters for each model are the same over all challenges. Details on these can be found in the supplement (*cf.* Sec. D). Unless specified otherwise, the evaluation datasets were sampled from the Michalski distribution. Finally, runs aborted due to code instabilities, memory overflows, or infinite loops are marked with ∗.

### 3.1 V-LoL Challenges

**Challenge 1: Visual Perception.** In the first challenge, we investigate the robustness of AI models to the visual perceptual challenges posed by V-LoL🚃 and V-LoL□. Here, we create two datasets based on the Theory X class rule on both visual representations, namely the train and the block representation. Fig. 4 presents the final test set accuracies of all AI models on both visualization types. It is evident that the evaluated models exhibit a remarkably similar level of performance on both.

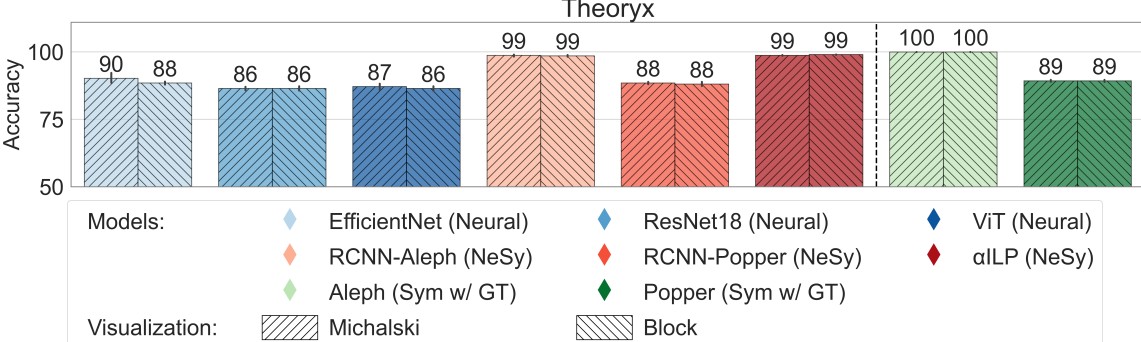

Figure 4: Visual Perception (Challenge 1). In this challenge, we compare the performance of various symbolic, neural, and neuro-symbolic AI models on different visual representations, i.e., the V-LoL⌑ and the V-LoL☐ datasets. Each bar depicts the average test accuracy along with a 95% confidence interval derived from a 5-fold cross-validation. We do not observe a strong influence of the different visual representations on the model's performances.

Alongside the different visual representations, V-LoL⌑ allows us to explore scene invariance. In Section C of the suppl., we conduct evaluations of the neural models across different background scenes. Throughout these evaluations (Fig. 11 and Tab. 4 and 5 (suppl.)) we can observe a modest decrease in accuracy as we transition from the base scene to more challenging scenes, such as desert, desert with sky, and finally, the fisheye scene. Furthermore, Tab. 5 (suppl.) in Sec. B.3 reveals that the neural models fail to generalize to unseen scenes. Specifically, across all three neural models, when tested on images from previously unencountered scenes, performance plummets to levels akin to random guessing. For the sake of brevity, all further experiments focus exclusively on the more train images set in the base scene.

**Challenge 2: Logical Reasoning.** In the second challenge, we specifically assess the models' performances in logical reasoning tasks. For this we create datasets of each of the logic rules described in Sec. 2.1. Fig. 5 presents the final results on each rule separately (additional results via LLM few-shot prompting can be found in Tab. 6 (suppl.)). It becomes evident that the nature of the logical problems has a pronounced impact on the performance of the models, leading to strong variations in their ability to handle the different logical class rules. While the neural approaches demonstrate a decent performance on Theory X, when confronted with the numerical and complex problems they are subject to a severe degradation in performance. This suggests that while these models can identify the attributes of objects, they struggle to understand and reason about numerical information, i.e. making arithmetic comparisons between different concepts, and performing long chains of non-trivial reasoning. Interestingly, $\alpha$ILP delivered the best performance on Theory X; however, when facing the numeric and complex problems the performance plummeted. In contrast, the Neuro|Symbolic AI systems, RCNN-Popper and RCNN-Aleph, demonstrate much greater abilities in dealing with the numerical problem. Although the purely symbolic AI methods perform even better, they are not truly comparable to the others, as they lack

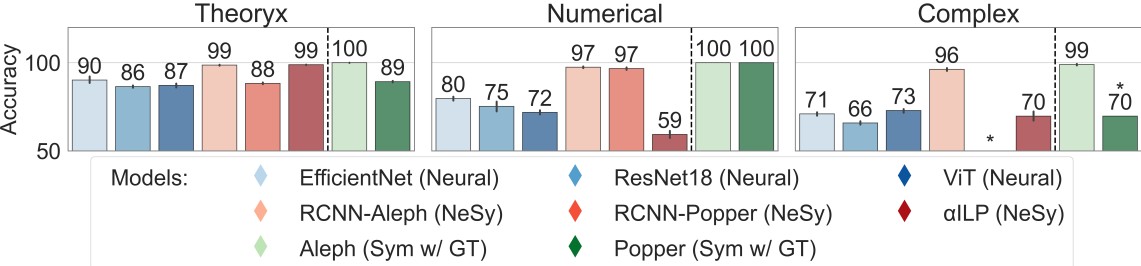

Figure 5: Logical Reasoning (Challenge 2). In this challenge, we compare the performance of various symbolic, neural, and neuro-symbolic AI models on the different "Theoryx", "Numerical", and "Complex" logic rules. Each bar depicts the average test accuracy along with a 95% confidence interval derived from a 5-fold cross-validation. Failed runs are denoted by an ∗. As the complexity of the logic rules increases (from the left to the right plot), a discernible decline in performance across all evaluated methods is observed.

the ability to handle the visual component of V-LoL⊟ in the first place. In this regard, the Neuro|Symbolic AI methods show a relatively robust behaviour with minor losses in performance over the different logical rules. However, RCNN-Popper failed to learn the complex problem due to endless loops and code instabilities during execution. Since Popper was able to fit on the symbolic ground truth of the dataset, this issue is probably caused by the combination of the rule complexity and perception noise caused by the RCNN module.

We provide further experiments in this context on the logical learning abilities of LLMs via zero-shot prompting in Sec. B.5 and Tab. 6 of the supplement. We observe that although the models, Llama2 and ChatGPT, successfully generate syntactically and semantically correct Prolog rules, both fail to learn logical features inherent in the input data.

In general, all models face their greatest challenge when dealing with the complex rule, emphasising the rule's intricacy and the advanced level of reasoning required to solve the task. Despite not being able to fully solve the individual logic task, RCNN-Aleph exhibits the best performance in inducing fairly reliable decision models across the logical problems. Remarkably, this mixed model that is in part based on a GOFAI system, Aleph, manages to outshine modern architectures such as the Vision Transformer and αILP in performance. However, in practice it suffers from poor optimization, resulting in prolonged runtimes and substantial memory usage.

**Challenge 3: Generalization.** One would assume that models that have truly solved the logical learning problems should be able to demonstrate this ability even on train compositions not seen during training.

In challenge 3 we assess models trained on V-LoL⊟ images encompassing up to four cars, using a test set featuring trains carrying 7 cars. The results in Fig. 6 expose limitations in the models' generalization abilities. Despite promising performances in the previous challenges, almost all models struggle to generalize effectively to longer train scenarios. This suggests a limited ability to adapt their reasoning capabilities to challenging, unseen inputs. Instead of comprehending the underlying problem and performing appropriate

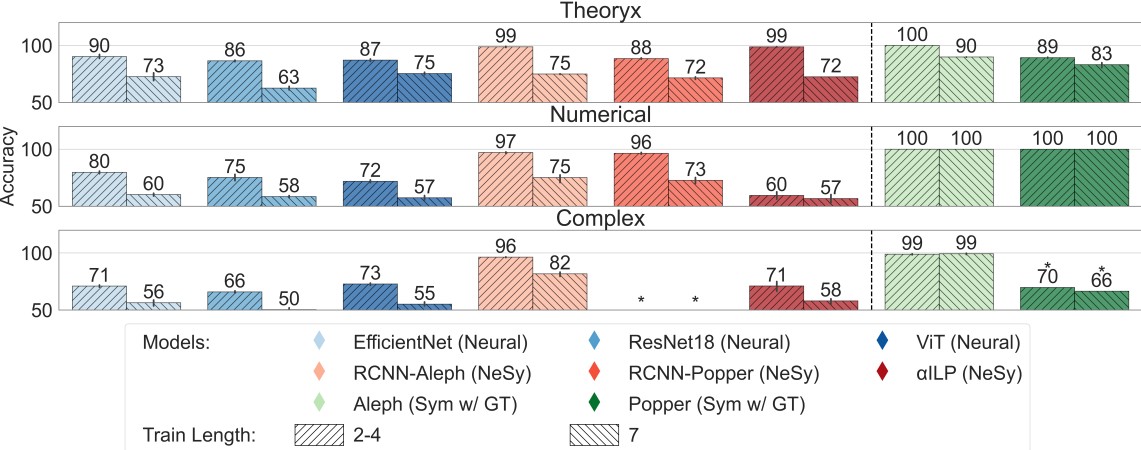

Figure 6: Generalization (Challenge 3). In this challenge, we investigate the generalization capabilities of symbolic, neural, and neuro-symbolic AI models when facing OOD V-LoL🚂 images that depict trains consisting of 7 cars. Each bar depicts the average test accuracy along with a 95% confidence interval derived from a 5-fold cross-validation. Failed runs are denoted by an ∗. We observe, that comparing the shorter train (2-4 cars) vs the longer train evaluations (7 cars) reveals strong performance degradations hinting at problems in terms of generalization over all models.

reasoning, the models tend to approximate the data distribution of the training set, leading to suboptimal generalization.

To test whether the neural models learn disentangled attribute representations, we conducted an additional evaluation using a test set in which train attributes were uniformly distributed, contrasting the Michalski distribution employed during training. The results of Fig. 12 in the supp. indicate that the neural models learn strong correlations between the attributes and lack the ability to generalize to out-of-distribution inputs. Specifically over all three models, when evaluated on images that stem from the uniformly distributed attribute distribution we observed a performance drop to close to random guessing.

**Challenge 4: Test-Time Interventions.** In challenge 4, we leverage the versatility provided by V-LoL🚂 that allows for performing test-time interventions on the composition of individual trains. We conduct two interventions on images of the Theory X challenge, providing us valuable insights into the model's reasoning process. Specifically, we investigate the models' class predictions of 2000 test images before and after performing the two different attribute interventions. We recall, Theory X trains have either a short, closed car, or a car with a barrel load is somewhere behind a car with a golden vase load. The first intervention, depicted in Fig. 7 (top left), assesses to which degree the models do learn the underlying spacial relationships. We initially evaluate the models on westbound trains that are carrying a barrel in front of a golden vase. Subsequently, we intervene by swapping barrel and golden vase payloads. This effectively alters the train's direction from west- to eastbound. Our second intervention explores the impact of redundant and class-irrelevant artefacts. We initially evaluate the models on eastbound trains that are carrying a golden

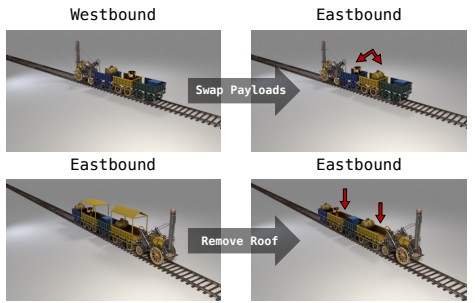

| V-LoL Interventions | Swap Pre-Swap → Post-Swap | Remove Pre-Remove → Post-Remove |
|---|---|---|
| ResNet18 | $85.41_{\pm 4.9} \to 30.02_{\pm 9.33}$ | $99.35_{\pm 0.42} \to 29.43_{\pm 11.85}$ |
| EfficientNet | $84.1_{\pm 7.16} \to 45.89_{\pm 24.08}$ | $99.67_{\pm 0.31} \to 49.36_{\pm 24.3}$ |
| ViT | $79.42_{\pm 4.28} \to 49.47_{\pm 13.21}$ | $98.91_{\pm 0.7} \to 51.09_{\pm 7.38}$ |
| RCNN-Aleph | $99.83_{\pm 0.38} \to 98.01_{\pm 3.29}$ | $99.82_{\pm 0.13} \to 100_{\pm 0}$ |
| RCNN-Popper | $90.5_{\pm 2.26} \to 10.98_{\pm 2.97}$ | $99.64_{\pm 0.15} \to 54.88_{\pm 14.68}$ |
| $\alpha$ILP | $100_{\pm 0} \to 100_{\pm 0}$ | $99.75_{\pm 0} \to 100_{\pm 0}$ |
| Aleph (GT) | $99.99_{\pm 0.02} \to 98.21_{\pm 3.4}$ | $100_{\pm 0} \to 100_{\pm 0}$ |
| Popper (GT) | $95.72_{\pm 1.87} \to 3.56_{\pm 1.77}$ | $100_{\pm 0} \to 36.14_{\pm 0.75}$ |

Figure 7: Interventions (challenge 4). In this challenge, we analyze the impact of test-time interventions on the classification performance of symbolic, neural, and neuro-symbolic AI models. The first intervention involves swapping payloads (left, top). The second involves removing roofs (left, bottom). The table (right) provides insights into each models' performance before and after intervention. These results indicate that while neural models exhibit challenges in adapting to the intervention, symbol-based AI demonstrate greater resilience under these test-time modifications.

vase in front of a barrel, with both of the cars having a closed roof. Subsequently, we intervene by removing all roofs from the trains, as illustrated in Fig. 7 (bottom left). The table in Fig. 7 (right) displays the models' classification accuracy before and after the interventions. While most models exhibit a decent performance on the original (pre-intervention) images, their accuracy significantly diminishes on the intervened images. Notably, neural models are easily fooled by the interventions. This suggests that they do not fully capture the underlying logical learning problem, either failing to grasp key special relationships or being swayed by redundant and class-irrelevant artefacts. In contrast, the neuro-symbolic models, RCNN-Aleph and $\alpha$ILP, appear to be less affected by these interventions.

**Challenge 5: Data-Efficiency.** Challenge 5 investigates the data efficiency of the individual models by assessing their performances across different training set sizes. The evaluation is conducted using varying numbers of training samples, namely 100, 1k, and 10k. As depicted in Fig. 8 in the small data regime, we observe notable variations in performances among the models. While the neural models are subject to severe performance degradation, the neuro-symbolic approaches showcase their ability to learn effectively also from small amounts of data. In particular, it can be observed that $\alpha$ILP achieves best performances on both Theory X and complex problems for 100 training samples. Intriguingly, these performances even surpass those achieved with 1k samples. Turning our attention to the large data regime (10k training samples), we note substantial performance improvements for neural approaches. On the other hand, the neuro-symbolic AI systems encounter challenges in effectively scaling their performances as $\alpha$ILP and Popper exhibit declines in performance. Popper additionally runs into endless loops and code instabilities during execution. Aleph, on the other hand, suffers from poor optimization. In our detailed runtime analysis of Aleph in Sec. B.6 of the suppl. we observe that an increase in the amount of data leads to an exponential training runtime and memory consumption. Notably, $\alpha$ILP consistently performs poorly on the numerical problem. This could potentially be attributed to issues with mode declaration and hyperparameter tuning. Comparing the individual

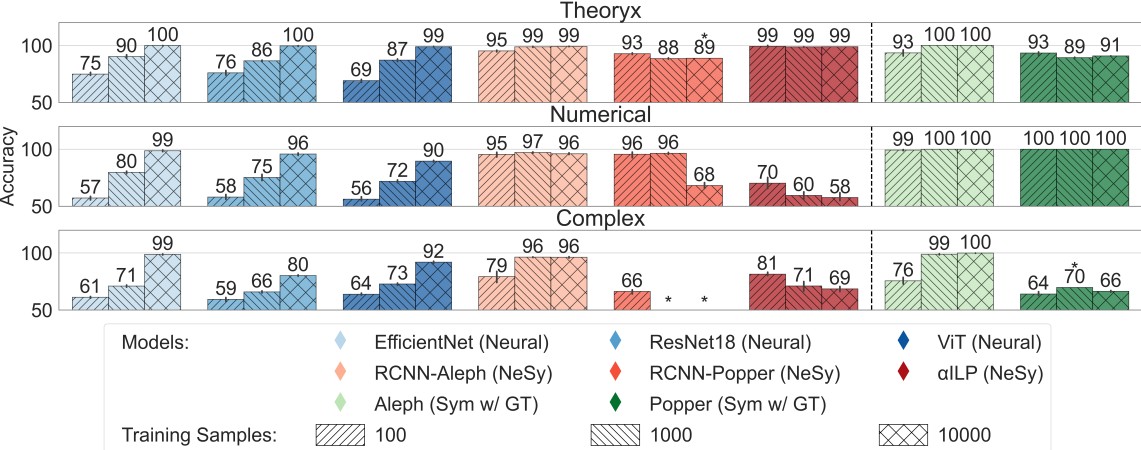

Figure 8: Data-Efficiency (Challenge 5). In this challenge, we evaluate the data-efficiency of various symbolic, neural, and neuro-symbolic AI models using V-LoL🔲. Each bar depicts the average test accuracy along with a 95% confidence interval derived from a 5-fold cross-validation. Failed runs are denoted by an ∗. While the neural models exhibit significant performance improvements with the access to large amounts of data, the NeSy methods face challenges with performance fluctuations and failed runs (e.g., due to exponential runtimes and code instabilities).

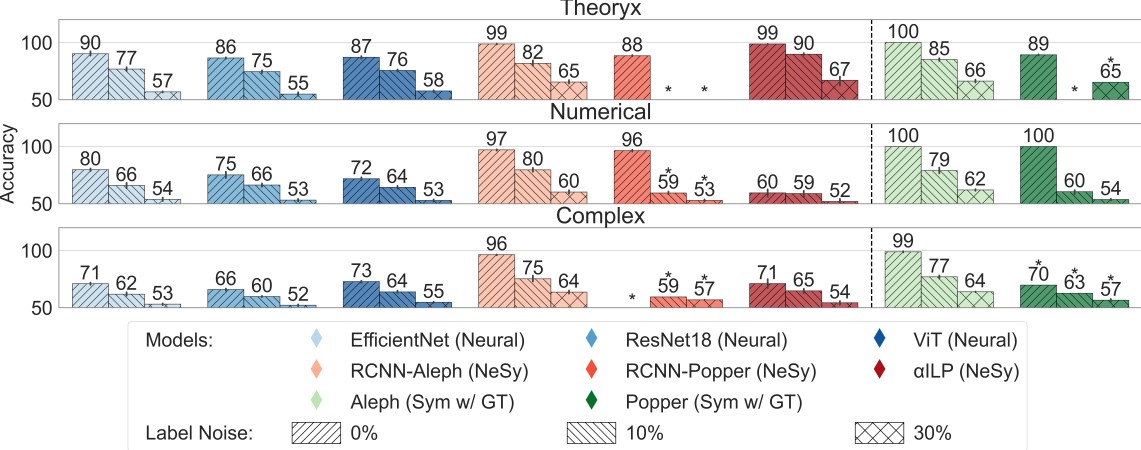

Figure 9: Label Noise (Challenge 6). In this challenge, we evaluate the robustness of the investigated AI models to varying degrees of label perturbations. Each bar depicts the average test accuracy along with a 95% confidence interval derived from a 5-fold cross-validation. Failed runs are denoted by an ∗. Especially, the ILP-based models, RCNN-Aleph and RCNN-Popper, struggle to cope with noise leading to severe performance losses and failed runs (e.g. due to prolonged run times, and code instabilities.

rules across the training size, it becomes evident that the complex rule generally poses the greatest challenge.

**Challenge 6: Label Noise.** In the final challenge, we investigate the robustness of the AI models to label noise. For this purpose, the respective AI systems are trained on datasets with specific amounts of perturbed labels (flipped labels). In Fig. 9, we compare the model's performances for different degrees of label perturbation. In general, all AI models have difficulties coping with the noise. In this experiment $\alpha$ILP seems to be least affected given the Theory X rule. On numerical and complex problems the ILP system Aleph maintains its lead position, although it also has to cope with large performance losses. Popper, however, is subject to strong turmoil that results in a performance close to random guessing and a multitude of failed runs. The investigated neural approaches all show a very similar level of degradation and are slightly less affected by the noise compared to the ILP ones.

## 3.2 Discussion

In our analysis we have evaluated and compared a range of AI architectures, and revealed several key shortcomings via learning challenges generated via V-LoL.

**Neural AI.** The investigated neural AI models show largely similar performances amongst themselves across the different challenges. While they can identify the attributes of objects, they struggle to reason about numerical information, e.g. making arithmetic comparisons between different concepts, as well as performing long chains of intricate reasoning (*cf.* challenge 2). These models tend to acquire pronounced biases from the training data, leading to challenges in overcoming these biases when confronted with data outside the training distribution (*cf.* challenge 3). The utilization of distinct data distributions during training and testing reveals a deficiency in achieving a proper disentanglement of object attributes in the learned representations of the neural models (*cf.* challenge 3). Challenge 4 particularly reveals that these models tend to be susceptible to small changes in input, being easily fooled by significant or insignificant perturbations in train compositions. Additionally, training on different dataset sizes exposes their reliance on a large training set size (*cf.* challenge 5).

**Symbolic AI.** While the symbolic AI models demonstrate strong performances across different visual logical learning problems (*e.g.*, challenge 2), it must be noted that they make a practically unrealistic assumption of omniscient perception. Furthermore, they are subject to certain challenges requiring excessive tuning of hyperparameters and priors (language biases), where even minor changes can lead to significant performance fluctuations in practice. Lastly, the application faces limitations that are particularly pronounced when dealing with noise or larger training samples (challenge 5 and 6) such as code instabilities, endless loops or excessive memory usage.

**Neuro-symbolic AI.** While neural AI struggles to perform visual logical reasoning, and symbolic AI lacks the ability to perceive and interact within a visual environment, neuro-symbolic AI emerges as a promising tool in the realm of visual logical learning. In the V-LoL-Challenges we can observe that neuro-symbolic AI predominantly outperforms purely neural AI in complex reasoning tasks, yet it is more susceptible to strong performance fluctuations. Although RCNN-Aleph is also not able to fully solve each individual V-

LoL-Challenges, it demonstrates the best overall performance, inducing the most reliable decision models across the different V-LoL-Challenges. Nonetheless, these models are not without limitations. Firstly, akin to the symbolic models, the neuro-symbolic models require significant tuning of hyperparameters and priors (language biases), where even minor changes can lead to strong performance fluctuations. Secondly, the noise arising from the perception module exacerbates the previously discussed optimization problems. Lastly, the translation process from visual input to a symbolic representation is inherently ambiguous. Accordingly, neuro-symbolic AI demands a stronger inductive bias, requiring additional background knowledge on the learning problem to mitigate the problem of ambiguity. For instance, one might view the train as a single entity or consider not only the number of wagons, payloads, and axles but also details like wheels, wooden bars, screws, and reflections or even their materials, positions, orientations, and relationships. In this context, additional background knowledge is needed to distill the relevant image features that align with the core aspects of the underlying learning problem. This, however, comes with both advantages and disadvantages. On the one hand, neuro-symbolic AI offers a more comprehensible decision-making process, granting the user insights and control over the image features used during learning. On the other hand, in cases where this background knowledge, i.e. the underlying learning problem, is unavailable or the link between classification and image content is unclear, defining an appropriate inductive bias becomes challenging. Despite these limitations, their abilities to flexibly handle the visual *and* logical reasoning components of the challenges makes them attractive approaches that lie between the purely neural and symbolic models.

**Key Takeaways.** The main goal of our evaluations was to illustrate the variety of challenges that can be instantiated via V-LoL and then utilized for identifying potentials and shortcomings of an AI approach. Despite the potential abilities of specific models for specific challenges, our findings reveal that all investigated approaches struggle to cope with the full set of the illustrated challenges. Intriguingly, RCNN-Aleph, an AI system rooted in the GOFAI principles, is able to outshine SOTA architectures such as $\alpha$ILP and Vision Transformer (*cf.* also preliminary LLM results in Sec. B.5). This surprising outcome sparks a discussion on the potential relevance and application of traditional AI systems in contemporary AI. Moreover, while existing datasets adhere to predetermined challenges for exploring different visual reasoning abilities — *e.g.*, Clevr-Hans (Stammer et al., 2021) addresses confounding data, PTR (Hong et al., 2021) focuses on relational learning in compositional objects — V-LoL, in contrast, offers a framework with a visual logical interface. This unique feature allows users to define their own visual logical learning challenges and create a dataset that enables the evaluation on self-defined problems beyond the ones of our evaluations. In comparison to existing datasets or subsets thereof V-LoL thus allows to iteratively establish and investigate multiple specifically tailored challenges while preserving easy comparability among these. In our experiments, we have lead the way and provided examples of such challenges, encompassing various complex visual and logical challenges, from test-time interventions to generalization challenges. These experiments not only enable us to compare AI approaches across the different AI methodologies — neural, symbolic, and neuro-symbolic AI — but also offer insights into respective methodologies thereby revealing both shortcomings and benefits that are inherent to these methodologies. V-LoL thus serves as a valuable

tool that grants AI researchers diagnostic insights into the visual logical reasoning processes of AI methods, and allows to suggest new avenues for exploration in future research.

**Limitations.** A primary limitation of our work lies in the synthetic nature of our dataset, which contrasts with the visual complexity of natural images. Future instantiations should investigate incorporating more naturally complex visual representations. However, this synthetic character allows for great versatility and generation power in terms of developing diagnostic challenges, *e.g.*, the direct access to the data generation process allows for easily performing test-time interventions. This represents an important property for developing targeted evaluations of AI models' visual logical learning abilities. A further limitation is that the current translation from visual to symbolic representations in V-LoL-Trains is unambiguous in the sense *e.g.*, that lighting conditions and occlusion can make it hard to identify objects and properties in real-world settings. Future instantiations should thus also explore visual scenes that reflect such natural uncertainties found in real-world settings. Lastly, care should be taken when developing datasets via the V-LoL framework in terms of potential dataset biases. Particularly, when investigating very complex logic rules it can occur that generated images contain unintended spurious correlations due to a reduced space of possible scene configurations. If these go unchecked it can be difficult to draw useful conclusions on a model's visual logical learning abilities.

## 4 Impact

The V-LoL dataset is related to datasets on visual reasoning from the field of DL, but also to symbolic AI benchmarks. Importantly, it has an impact on various subfields of AI.

**Visual (reasoning) datasets.** The transition from purely visual perception tasks to visual reasoning has led to the development of specialized datasets that challenge AI models to perform different reasoning tasks based on visual information. These datasets incorporate tasks such as spatial, relational, temporal, analogical and causal reasoning, providing a more comprehensive evaluation of AI systems' cognitive abilities beyond simple perception tasks. Notable examples include VQA (Antol et al., 2015; Wu et al., 2017; Krishna et al., 2017; Johnson et al., 2017), QAR by Huang et al. (2021), CLEVRER by Yi et al. (2020), CLEVR-Hans by Stammer et al. (2021), MNIST-Addition by Manhaeve et al. (2018), GQA dataset by Hudson and Manning (2019). Also more cognitively inspired datasets such as the PTR dataset by Hong et al. (2021), the RAVEN dataset by Carpenter et al. (1990) or the datasets of Webb et al. (2021) and Kerg et al. (2022). Lastly, datasets of Gopnik and Sobel (2000) and Zhang et al. (2021), focusing on causal-based visual reasoning, and game-based datasets for concept learning by Bramley et al. (2018). All of these have paved the way for advancing visual reasoning research in the field of computer vision. A comparison of a selection of datasets, their features and learning tasks can be found in Tab. 1. While most of these tasks assume pre-defined programs to compute answers, V-LoL takes a different approach by requiring agents to learn abstract logic programs for classification, adding a unique dimension to the field of visual reasoning.

**Classical ILP benchmarks.** ILP benchmarks have been an integral part of the AI field since its inception to evaluate AI systems' performance in logical learning and knowledge representation. These benchmarks involve learning logical rules or programs from examples and background knowledge, encompassing challenges such as logical inference, relational reasoning, and generalization from limited examples. Examples of popular ILP benchmarks include the Michalski train problem (Michalski, 1980; Michie et al., 1994), Bongard Problems (Bongard, 1968), Kinship (Dua and Graff, 2017), Mutagenesis (Debnath et al., 1991), and Bongard-LOGO (Nie et al., 2020).

**AI systems.** V-LoL has the distinct advantage of allowing evaluation and comparison of AI systems from the domains of symbolic, neural, and neuro-symbolic AI. Symbolic AI, utilizing representations like First-Order Logic (FOL), provides essential knowledge representation and reasoning capabilities (Baral, 2010; Brachman and Levesque, 2004; Nickel et al., 2015). ILP has been established as a technique to learn generalized rules using FOL as its language (Muggleton, 1995; Nienhuys-Cheng et al., 1997; De Raedt and Kersting, 2008; Cropper et al., 2020), offering advantages such as learning explicit programs and learning from small data. Deep Learning, a prominent technique in neural AI, has shown significant achievements in various AI tasks (LeCun et al., 2015; Silver et al., 2016; Jumper et al., 2021), although lacking interpretable and explainable reasoning steps. The emerging field of neuro-symbolic AI integrates symbolic computations and neural networks (Garcez and Lamb, 2023), enabling efficient parameter estimation (*e.g.*, DeepProbLog (Manhaeve et al., 2018), NeurASP (Yang et al., 2020), SLASH (Skryagin et al., 2022), NS-CL (Mao et al., 2019), and differentiable theorem provers (Rocktäschel and Riedel, 2017)) and explicit logic program learning from raw data (*e.g.*, $\partial$ILP (Evans and Grefenstette, 2018), $\alpha$ILP (Shindo et al., 2023), and FFNSL (Cunnington et al., 2023)). V-LoL serves as a diagnostic benchmark for evaluating and advancing AI systems across all of these different domains.

## 5 Conclusion

We have introduced V-LoL, a dataset specifically designed for the diagnostic evaluation of visual logical learning in AI models. Specifically, V-LoL⊞⌐, an initial instantiation of V-LoL, provides a versatile generator that allows to generate custom datasets for detailed investigations of the capabilities and limitations of current and future AI models. In this way it is possible to investigate various learning abilities of AI models, ranging from perception and relational reasoning to arithmetic reasoning and test-set generalization. V-LoL⊞⌐ thus offers the ability to finely adjust the learning task, enabling a comprehensive *diagnostic* analysis of various AI models for visual logical learning that goes beyond *benchmark* datasets, but also previous diagnostic datasets. Specifically, we provide diagnostic evaluations of several symbolic, neural, and neuro-symbolic models and could identify key shortcomings and benefits arising from the individual methodologies. One finding from this is that models based on classical AI approaches (*e.g.*, Aleph), overall show promising properties over SOTA models, though requiring a stronger inductive bias in form of background knowledge. Overall, by merging logic and vision, V-LoL ultimately poses an attractive tool for ongoing

research efforts aimed at enhancing the performance and capabilities of AI models, and further driving progress in developing models with capabilities for visual logical learning.

Future evaluations include assessing the visual logical learning capabilities of large-scale vision-language. Investigating the models' generalization across different background scenes can provide crucial information on their robustness to out-of-distribution data. Furthermore, conducting extensive human evaluations will provide unique perspectives on the challenges posed by V-LoL. Finally, extending V-LoL to include other ILP problems (*e.g.*, Bongard Bongard (1968), RoboCup Kitano et al. (1997), Poker Blockeel et al. (1999) or MutaGenisis Srinivasan et al. (1996)), but also moving to the domain of 3D image sequences are important directions for future research.

**Ethical Statement.** The V-LoL dataset is a diagnostic dataset aimed at explicitly investigating various challenges of visual logical learning and in this way is designed to identify the capabilities and potential shortcomings of AI models. It can, however, also serve as a benchmark for improving AI models in general. Therefore, although it is not its primary intention, providing such a benchmark can also have the effect *e.g.*, of improving models that are to be used in a harmful way. Overall, the deployment of visual logical AI models particularly in high stakes scenarios such as autonomous driving or medical diagnosis should be carefully checked to avoid potentially fatal predictions. However, the particular goal of V-LoL is to be able to identify any model shortcomings based on specifically tailored evaluations and analyses. Lastly, as the introduced datasets are synthetically generated no harm was done in the generation of the dataset.

## Acknowledgments

The authors thank the reviewers and action editor for their valuable feedback and guidance, which helped improve the quality of this paper. This work has benefited from the HMWK project "The Third Wave of Artificial Intelligence - 3AI", Hessian.AI, and the Hessian research priority program LOEWE within the project "WhiteBox". Further, we acknowledge support of the hessian.AISC Service Center (funded by the Federal Ministry of Education and Research, BMBF, grant No 01IS22091), and the EU ICT-48 Network of AI Research Excellence Center "TAILOR" (EU Horizon 2020, GA No 952215). The Eindhoven University of Technology authors received support from their Department of Mathematics and Computer Science and the Eindhoven Artificial Intelligence Systems Institute.

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

# Supplemental Materials

## A V-LoL dataset

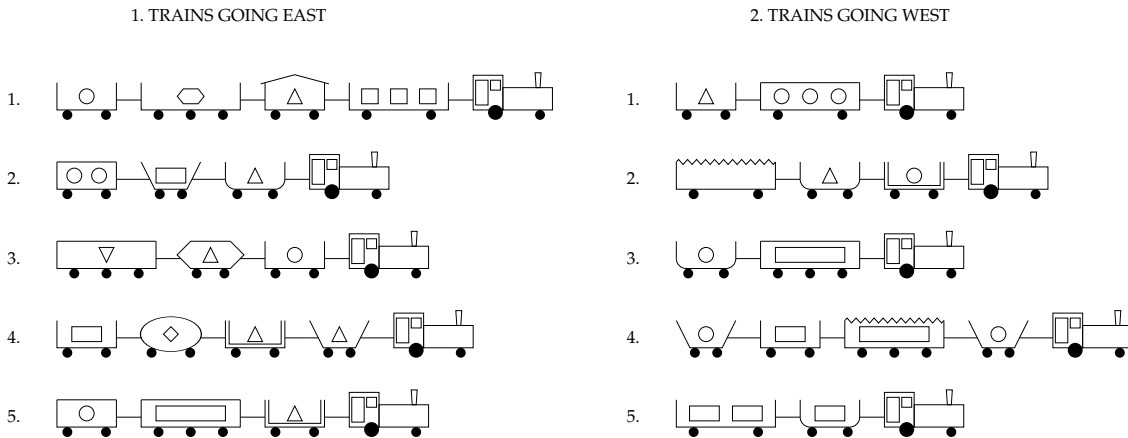

Figure 10: Michalski's original set of trains Michalski (1980)

### A.1 Details on Michalski Train Semantics

E. Bloedorn et al.Bloedorn et al. (1995) state, that Muggleton's Prolog representation shows subtle differences to the original representation from Michalski such that artefacts are created which are not present in Michalski's original work. These artefacts result from an ambiguous definition of the individual descriptors. Consequently, depicted cars without a load must be assigned a description of the load shape, even though the number of loads remains zero. In this sense, any load shape can be assigned to a car without visually affecting the depicted car. This is an inconsistency that is not present in the original Michalski trains. However, as the images contain unambiguous information about the train, it is not present in the visual V-LoL classification task. Nevertheless, it is beneficial to remove this inconsistency in the background knowledge of the dataset. In V-LoL we circumvent this problem by introducing the value "none" for the load shape descriptor.

Trains sampled from the random distribution can be generated in one of 23.4 trillion combinations, assuming a train length of 2 - 4 cars. This number of permutations grows exponentially with every additional car.

### A.2 Attribute Constraints

In the following we list the attribute constraints that are enforced via Muggleton's Muggleton (1998) Prolog train generator:

1. A train has two, three or four cars, each of which can either be long or short.

Table 2: Attributes of Michalski's original train attributes. The Table gives an overview of the original assignable values of each descriptor. For the respective descriptors, the above-mentioned interrelationships must be taken into account such that some attributes might be mutually exclusive.

| Car Position | Car Shape | Car Length | Car Wall | Car Roof | Wheels Num. | Load Num. | Load Shape |
|---|---|---|---|---|---|---|---|
| 1 | Rectangle | Short | Single | None | 2 | 0 | Rectangle |
| 2 | Bucket | Long | Double | Arc | 3 | 1 | Triangle |
| 3 | Ellipse | | | Flat | | 2 | Circle |
| 4 | Hexagon | | | Jagged | | 3 | Diamond |
| | U shaped | | | Peaked | | | Hexagon |
| | | | | | | | U-triangle |

2. A long car can have either two or three axles.

3. A short car can be rectangular, u-shaped, bucket-shaped, hexagonal, or elliptical, while a long car must be rectangular.

4. A hexagonal or elliptical car is necessarily closed, while any other car can be either open or closed.

5. The roof of a long-closed car can be either flat or jagged.

6. The roof of a hexagonal car is necessarily flat, while the roof of an elliptical car is necessarily an arc. Any other short closed car can have either a flat or a peaked roof.

7. If a short car is rectangular then it can also be double-sided.

8. A long car can be empty or it can contain one, two or three replicas of one of the following kinds of load: circle, inverted-triangle, hexagon, or rectangle.

9. A short car contains either one or two replicas of the following kinds of load: circle, triangle, rectangle, or diamond.

10. No sub-distinctions are drawn among rectangular loads, even though some are drawn square and others more or less oblong. The presumption is that they are drawn just as oblong as they need to be in each case to fill the available container space.

11. In Michalski's original version a possible distinction between hollow and solid wheels was ignored, as also here.

## A.3 Prolog and FOL Notation of the Classification rules

In the following we provide the Prolog and first-order-logic (FOL) representations of the Theory X, Numerical and Complex logic rules that were used in our evaluations.

Theory X is defined as follows:

$$eastbound(Train) \models \exists Car_1, Car_2,$$

$$has\text{-}car(Train, Car_1) \wedge has\text{-}car(Train, Car_2)$$
$$\wedge \left( (short(Car_1) \wedge closed(Car_1)) \right.$$
$$\vee \left( has\text{-}load(Car_1, golden\text{-}vase) \right.$$
$$\wedge has\text{-}load(Car_2, barrel)$$
$$\wedge somewhere\text{-}behind(Train, Car_2, Car_1)))$$

Prolog:

```prolog
eastbound([Car|Cars]):- (short(Car), closed(Car));
(has_load0(Car,triangle),has_load1(Cars,circle));
eastbound(Cars).
```

The numerical rule is defined as follows:

$$eastbound(Train) \models \exists Car_1 \; has\text{-}car(Train, Car_1) \wedge load\text{-}num(Car_1, N)$$
$$\wedge car\text{-}num(Car_1, N) \wedge has\text{-}wheel(Car_1, N)$$

Prolog:

```prolog
eastbound(Train):- has_car(Train,Car),load_num(Car,N), car_num(Car,N),
has_wheel0(Car,N).
```

The complex rule is defined as follows:

$$eastbound(Train) \models \exists Car_1, Car_2, Car_3$$
$$has\text{-}car(Train, Car_1) \wedge (has\text{-}car(Train, Car_2) \wedge has\text{-}car(Train, Car_3) \wedge$$
$$(load\text{-}num(Car_1, N1) \wedge car\text{-}num(Car_1, N2) \wedge$$
$$has\text{-}wheel0(Car_1, N3) \wedge (N2 < N1) \wedge (N2 < N3))$$
$$\vee (short(Car_1) \wedge long(Car_2) \wedge car\text{-}num(Car_1, N1) \wedge car\text{-}color(Car_1, A) \wedge$$
$$car\text{-}color(Car_2, A) \wedge has\text{-}wheel(Car_2, N2) \wedge (N1 < N2))$$
$$\vee (car\text{-}color(Car_1, X) \wedge car\text{-}color(Car_2, Y) \wedge car\text{-}color(Car_3 \wedge Z) \wedge$$
$$\neg(X = Y) \wedge \neg(Y = Z) \wedge \neg(Z = X))$$

Prolog:

```prolog
eastbound(Train):-
has_car(Train,Car1),has_car(Train,Car2), has_car(Train,Car3),
(load_num(Car1,N1), car_num(Car1,N2),has_wheel0(Car1,N3), N2 < N1, N2 < N3;
short(Car1), long(Car2),car_num(Car1,N1), car_color(Car1, A),
car_color(Car2, A),has_wheel0(Car2,N2), N1 < N2;
car_color(Car1,X), car_color(Car2,Y),car_color(Car3,Z),X/=Y, Y/=Z, Z/=X).
```

## A.4 Reasoning Properties for Logic Rules

Successfully solving the individual V-LoL challenges requires a set of learning and reasoning abilities, including:

- **Object recognition** is a crucial component of visual reasoning and involves identifying and categorizing objects based on their physical attributes.

- **Counting** is also an essential skill in the domain of visual reasoning, which involves determining and understanding the number of objects or occurrences of a particular feature. In the case of V-LoL⌨ it is required to accurately count the number of occurrences of objects and concepts such as the number of cars, payloads or car axles.

- **Relational reasoning** is another essential type of reasoning which involves understanding and drawing conclusions based on relationships between multiple objects and concepts. For example, understanding that a barrel load is always located in a car in front of a car with a golden vase load.

- **Spatial reasoning** involves drawing conclusions based on the spatial information of individual objects within a scene. E.g. in V-LoL⌨ it is required to understand and conclude which car is in front of another in the direction of travel, i.e. in the direction of the locomotive.

- **Arithmetic reasoning** involves reasoning based on numerical information. Therefore, it involves understanding arithmetic operators and comparisons such as ($<, >, \neq$, $==$). For example, it is important to understand whether one car has less payloads than another.

- **Analogical reasoning** involves drawing conclusions based on analoguos relationships between components of individual objects. In V-LoL⌨, it is required to find analogical relationships in the attributes of the individual cars. E.g. all yellow cars have the same payload.

- **Abstract reasoning** is the ability to draw conclusions based on abstract ideas, concepts, and patterns that are not concretely tied to specific objects (immediately apparent). For example, drawing conclusions by comparing the number of axles with the position of a car or the number of payloads requires abstract reasoning skills.

- **Exact logical reasoning** is the ability to understand and reason based on logical relations which involves understanding and concluding based on logical operators (e.g. $\equiv, \wedge, \vee, \neg$). Logical reasoning is essential for V-LoL⌨ since it involves solving complex reasoning problems that constitute multiple logical operators.

An overview on which class rule (Theory X, Numerical and Complex) addresses which of the visual logical abilities can be found in Tab. 3 showing that each rule allows to investigate a different subset of V-LoL abilities.

## A.5 Details on V-LoL Train Semantics

Specifically, the trains of V-LoL⌨ consist of a fixed locomotive object pulling a variable number of train cars. Each car is assigned one of five colours: *yellow*, *green*, *grey*, *red*, or *blue.* Moreover, cars exhibit different roof styles, including a empty roof *frame*, a *flat* roof, a *barred* roof, a *peaked* roof, or *no roof* at all. The walls of a car can feature either a *solid*

Table 3: The V-LoL-Trains problems and the visual logical learning challenges that they address individually. We here categorize based on *object recognition* and *counting* abilities and *relational*, *spatial*, *analogical*, *arithmetic*, *abstract* and *exact* logical reasoning. For each of our three V-LoL-Trains classification challenges, "Theory X", "Numerical", and "Complex" we indicate which of these reasoning skills are required to solve the tasks.

| V-LoL⬚ challenges | Object recognition | Counting | Relational | Spatial | Analogical | Arithmetic | Abstract | Exact Logic |
|---|---|---|---|---|---|---|---|---|
| Theory X | ✓ | ✗ | ✓ | ✓ | ✓ | ✗ | ✗ | ✓ |
| Numerical | ✓ | ✓ | ✓ | ✓ | ✗ | ✓ | ✓ | ✓ |
| Complex | ✓ | ✓ | ✓ | ✓ | ✓ | ✓ | ✓ | ✓ |

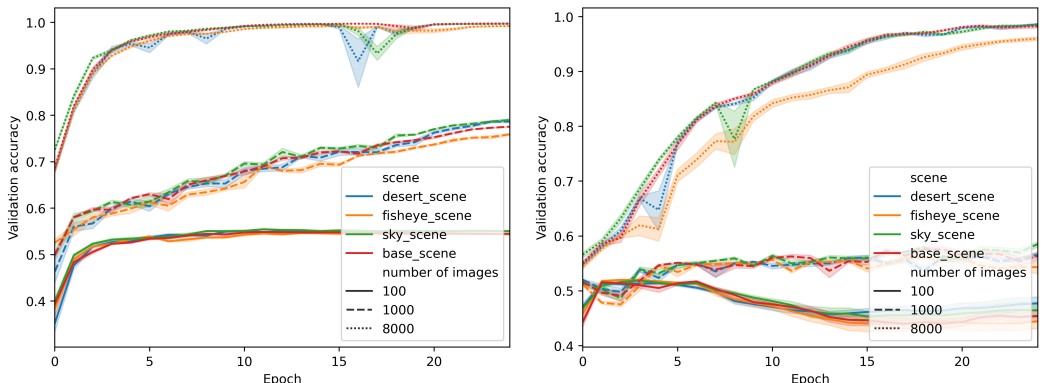

Figure 11: 5-fold cross-validation on the perceptual performance of the ResNet-18 (left) and Mask-RCNN (right) models. The models are evaluated across the four different scenes from the V-LoL⬚ dataset and varying quantities of training images. For each validation, a separate set of 2,000 images, excluded from training, is used. The displayed graphs highlight the average accuracy for detecting the train's attributes, complemented by their respective confidence intervals (CI).

wall or *railings*. Additionally, each car can either be *long* or *short*, equipped with either two or three axles, and capable of carrying a minimum of zero and a maximum of three loads. The available load types include *blue boxes*, *golden vases*, *barrels*, *diamonds*, *metal pots*, and *oval vases*.

## B Additional experiments

### B.1 Neural Perceptual Performance Across Different Scenes

In this experiment, we explore the impact of various background scenes provided by the V-LoL⬚ dataset on the perceptual performance of neural networks. To achieve this, we conduct evaluations using both a ResNet-18 and a Mask-RCNN model to predict individual attributes of the V-LoL⬚ within the images. The models are trained and evaluated on the same background scenes. The performance results, depicted in Fig. 11, portray the outcomes of the ResNet-18 (left) and Mask-RCNN (right) models across the different scenes. Across

Table 4: Average test accuracy and the respective standard deviation on the V-LoL⧉ challenge. The neural models are trained on different background scenes with varying training set sizes.

| ResNet | | | |
|---|---|---|---|
| | **100 images** | **1000 images** | **8000 images** |
| **Base scene** | 75,15 ± 3,04 | 87,32 ± 1,62 | 99,63 ± 0,14 |
| **Desert scene** | 76,66 ± 1,76 | 88,58 ± 1,22 | 99,32 ± 0,19 |
| **Sky scene** | 76,78 ± 3,11 | 87,86 ± 0,87 | 98,77 ± 0,71 |
| **Fisheye scene** | 74,21 ± 4,4 | 85,93 ± 1,34 | 98,96 ± 0,32 |
| EfficientNet | | | |
| | **100 images** | **1000 images** | **8000 images** |
| **Base scene** | 74,41 ± 3,83 | 90,43 ± 1,83 | 99,86 ± 0,08 |
| **Desert scene** | 73,20 ± 1,88 | 90,14 ± 1,84 | 99,66 ± 0,12 |
| **Sky scene** | 72,70 ± 2,74 | 91,45 ± 0,85 | 99,52 ± 0,15 |
| **Fisheye scene** | 70,36 ± 2,17 | 88,33 ± 0,91 | 99,47 ± 0,21 |
| Vision Transformer | | | |
| | **100 images** | **1000 images** | **8000 images** |
| **Base scene** | 63,11 ± 7,28 | 84,40 ± 2,65 | 97,17 ± 0,36 |
| **Desert scene** | 62,81 ± 8,24 | 71,26 ± 4,39 | 92,40 ± 3,71 |
| **Sky scene** | 60,01 ± 4,29 | 81,97 ± 2,83 | 93,03 ± 4,12 |
| **Fisheye scene** | 64,58 ± 3,30 | 75,88 ± 3,47 | 92,74 ± 2,39 |

both models, we discern a decline in accuracy as we transition from the base scene to more challenging scenes, such as desert, desert with sky, and finally, the fisheye background. We also show how varying the sample sizes available during training (i.e., 100, 1000, and 8000 samples) affect the models' accuracy. We observe that with smaller sample sizes (100 and 1000), the models' performance remains unsatisfactory. With an expanded training set comprising 8000 samples, there is a notable enhancement in perceptual performance; nevertheless, perfect perception is still not attained, especially in the case of more intricate scenes. These findings underscore the models' sensitivity to scene complexity and available training data.

## B.2 Neural Classification Performance Across Different Scenes

In this evaluation we scrutinize the classification performance of neural networks across the different background scenes provided by V-LoL⧉. We employ the three nerual models, namely the ResNet18, EfficientNet, and the Vision Transformer on the V-LoL⧉ TheoryX challenge. Our approach involves training these models with various sample sizes and evaluating their performance on a distinct holdout test set corresponding to the same scene. The results are summarized in Tab. 4. We can indeed observe differences in performances across the three models, with a decrease in accuracies from the base scene, through desert and desert with sky, culminating in the fisheye scenario which mostly yields the lowest accuracy.

Table 5: Average test accuracy and the respective standard deviation of the neural models evaluated on OOD scenes. The models are trained on the base scene with varying training set sizes. The evaluation is performed on 10.000 images of the other scenes. The baseline represents the initial validation performance of the models averaged across the three different scenes.

| ResNet | | | |
|---|---|---|---|
| | **100 images** | **1000 images** | **8000 images** |
| **baseline** | 75,88 ± 3,09 | 87,46 ± 1,14 | 99,02 ± 0,41 |
| **desert scene** | 69,85 ± 6,84 | 68,84 ± 11,81 | 83,13 ± 3,06 |
| **sky scene** | 67,82 ± 7,39 | 72,04 ± 8,05 | 83,14 ± 2,47 |
| **fisheye scene** | 63,97 ± 7,07 | 65,64 ± 7,09 | 76,88 ± 0,80 |
| EfficientNet | | | |
| | **100 images** | **1000 images** | **8000 images** |
| **baseline** | 72,09 ± 2,27 | 89,97 ± 1,20 | 99,55 ± 0,16 |
| **desert scene** | 52,13 ± 2,85 | 52,72 ± 5,60 | 60,88 ± 8,29 |
| **sky scene** | 54,20 ± 0,72 | 50,71 ± 2,71 | 55,92 ± 6,02 |
| **fisheye scene** | 53,67 ± 0,30 | 50,54 ± 3,52 | 56,29 ± 4,58 |
| Vision Transformer | | | |
| | **100 images** | **1000 images** | **8000 images** |
| **baseline** | 62,47 ± 5,28 | 76,37 ± 3,56 | 92,72 ± 3,40 |
| **desert scene** | 60,34 ± 10,96 | 72,85 ± 5,26 | 89,70 ± 0,99 |
| **sky scene** | 62,36 ± 11,69 | 67,71 ± 5,04 | 80,04 ± 2,89 |
| **fisheye scene** | 59,90 ± 8,53 | 67,52 ± 5,22 | 73,54 ± 3,31 |

### B.3 Out-of-Distribution: Classification Performance Across Different Scenes

In this section, we delve into the classification performance of neural models across the different background scenes provided by the V-LoL⬚ dataset. However, unlike the previous experiment, we focus on handling the challenge of OOD input. This is achieved by training the models on the base scene and evaluating their performance on the previously unobserved scenes of V-LoL⬚. The findings presented in Table 5 illustrate the significant decline in model performance when subjected to OOD scenarios. This discernible lack of robustness in the face of inconsequential variations in the background suggests that the model struggles to generalize beyond the confines of the training distribution.

### B.4 Out-of-Distribution: Different Attribute Distributions

In this evaluation we investigate the effect of performing inference on V-LoL⬚ images with an attribute distribution that is different from that observed during training. Specifically, we investigate the three neural models that were trained on the Michalski attribute distribution, but evaluated on the randomly distributed attributes. The images were generated with the Theory X rule.

Fig. 12 presents the results, where we provide the accuracies of the three models that were trained on 100 (small data regime), 1000 (medium data regime) and 10000 images

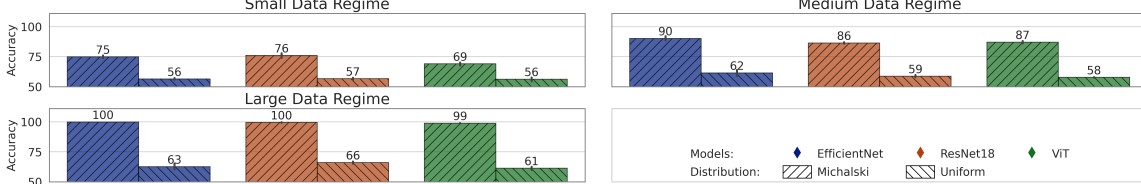

Figure 12: Out-of-Distribution Evaluation. Models were trained on images that were generated with the Michalski attribute distribution and evaluated on two test sets: one also generated via the Michalski distribution and one generated via the random (uniform) attribute distribution. We also provide results here for three training set sizes: 100 (small data regime), 1000 (medium data regime) and 10 000 training images (large data regime).

Table 6: Performance Metrics of LLMs on the logical learning challenges of V-LoL🚃. The table presents the test accuracy and standard deviation from 5-fold cross-validation across the three logical learning challenges: TheoryX, Numerical, and Complex. The number of 🚃 samples represent the number of train samples provided in each prompt. Entries marked with a '-' indicate that all runs are unsuccessful, while entries marked with '*' indicates single runs that failed. These failures were due to induced prolog rules that were syntactically or semantically invalid or because the input exceeded the model's maximum input length.

| Challenges | TheoryX | Numerical | Complex |
| 🚃 Samples | 8 / 20 | 8 / 20 | 8 / 20 |
| --- | --- | --- | --- |
| Llama2-13b | $49.07 \pm_{1.86}$ / $-$ | $49.99 \pm_{0.01}$ / $-$ | $49.61 * \pm_{1.34}$ / $-$ |
| Llama2-70b | $50.0 \pm_{0.0}$ /$-$ | $50.0 \pm_{0.01}$ / $-$ | $50.0 \pm_{0.0}$ / $-$ |
| ChatGPT-3.5 | $52.09 \pm_{3.62}$ / $50.73 \pm_{3.14}$ | $50.0 \pm_{0.01}$ / $51.47 \pm_{2.21}$ | $50.22 \pm_{0.35}$ / $51.42 \pm_{2.04}$ |
| ChatGPT-4 | $50.78 \pm_{1.25}$ / $-$ | $51.02 \pm_{2.47}$ / $-$ | $50.44 \pm_{0.88}$ / $-$ |

(large data regime) and tested on the test set that was generated with the Michalski attribute distribution as well as on the test set that was generated with the random attribute distribution. We can observe a strong decrease in model performance when evaluated with the out-of-distribution test set. This decrease can be observed in all data size regimes, but becomes even larger the larger the training set size becomes. These results suggest that all three investigated neural models tend to overfit to the training distribution for solving the reasoning tasks.

## B.5 Zero-shot Prompting with Large Language Models

In this section we evaluate LLMs on the V-LoL🚃 challenge using zero shot prompting. In the individual prompts the LLM are provided with a number of ground truth descriptions of set of labeled trains and tasked to induce a decision hypothesis in the prolog description language that is able to separate the provided trains. The number of provided train samples was chosen depending on the maximum input length the individual models can handle with.

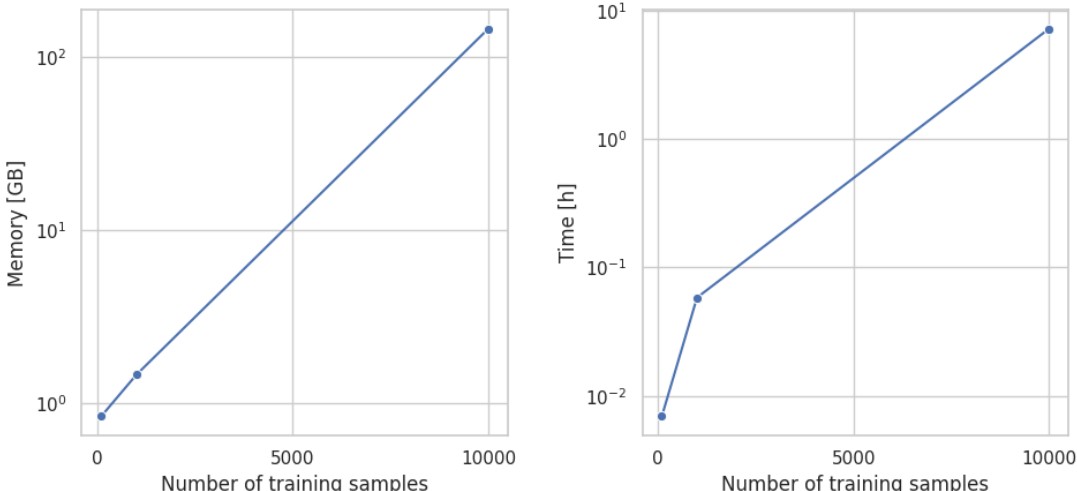

Figure 13: Runtime metrics for Aleph collected in Challenge 5: Data Efficieny. Moving from small to larger amounts of data we can observe an exponential increase in memory and time consumption.

Adding further samples have shown to result in a semantic and syntactic degradation of the induced decision rules.

Here we investigate the three different logical challenges, namely TheoryX, Numerical, and Complex. Over all experiments the LLMs are only able to achieve a accuracy of around 50% which is equivalent to random guessing. However most of the induced decision hypothesis, while not being logically correct, they at least where semantically and syntactically correct. The results indicate that SOTA LLMs still have strong difficulties recognising patterns in the input prompts, finding combinatorial relations and performing logical reasoning.

## B.6 Runtime analysis of Aleph on Challenge 5: Data Efficieny

In this section we provide a further analysis of runtime metrics, specifically focusing on the memory and time consumption that is required by Aleph in the context of Challenge 5: Data Efficiency. Fig 13 reveals an exponential growth in both runtime and memory as the scale of training data increases.

## C Implementation details

For Popper we set the hyperparameters to allow for a maximum of 10 rules each allowing a maximum of 6 variables and 6 literals in its body. Predicate finding and recursion are turned off, as we could not observe any performance improvement. For ALEPH we use the following hyperparameters: $clauselength = 10$, $minacc = 0.6$, $minscore = 3$, $minpos = 3$,

$nodes = 5000$, $explore = true$, $max\_features = 10$. Both ILP systems are trained and evaluated on the a symbolic ground-truths instead of the visual images.

Our subsymbolic models, namely the ResNet, EfficientNet, and Vision Transformer are initialized with the weights of the pre-trained foundation models which was trained on the 1000-class ImageNet dataset. The last fully-connected layer is replaced to fit the two-class classification task of westbound and eastbound trains. Subsequently, the models are transfer trained on the respective datasets for 25 epochs using a batch size of 50 and starting with a learning rate of 0.001 (0.0001 for the Vision Transformer), which decreases by 20% every five epochs. The Adam optimizer is used for updating the models' weights and the cross-entropy loss function for calculating the loss.

For the perceptions modules of the Neuro-Symbolic AI systems, we modify the improved mask-RCNN (v2 version) Li et al. (2021) to allow for multi-label instance segmentation. For more in depth implementation details please refer to our code. We initialize our model with pre-trained weights for MaskRCNN + ResNet50 + FPN using the v2 variant with post-paper optimizations. We transfer train our model on 10k V-LoL dataset containing random trains. For training we use the AdamW optimizer and cross-entropy loss. After inferring the segmented masked using mask-RCNN we post process these using a mask matching algorithm to assemble a symbolic scene representations. We achieve nearly 100% validation accuracy on the random V-LoL and 99% test accuracy on the Michalski V-LoL. Subsequently, we fit the ILP approaches using the same hyperparameters as in the run of the purely symbolic AI systems. For beam search of $\alpha$ILP we choose a beam size of 70 with a beam depth of 5. We select a maximum of 1000 clauses after search on which we then perform learning for 100 epochs. For TheoryX and the numerical rule we learn a logic program consisting of two rules while for the complex rule we learn 4 rules. For more in depth information on the mode declaration and hyper parameters of $\alpha$ILP, Popper, and Aleph please refer to our code.

All code was run on multiple NVIDIA A100-SXM4-40GB gpus.

## D Dataset and Code Availability and License

Access to the data set, data generator and experimental code for reproducing the results are bundled on our website: `https://sites.google.com/view/v-lol`. All data is released under the Creative Commons CC BY 4.0 license. All code is released under the MIT license.

## E Dataset Documentation: Datasheets for Datasets

Here we answer the questions outlined in the datasheets for datasets paper by Gebru et al.Gebru et al. (2021).

### E.1 Motivation

**For what purpose was the dataset created?** V-LoL was created to serve as a challenging benchmark for visual logical learning. By incorporating intricate visual scenes and flexible logical reasoning tasks within a versatile11 framework, V-LoL provides a platform for investigating a wide range of visual logical learning challenges.

**Who created the dataset (e.g., which team, research group) and on behalf of which entity (e.g., company, institution, organisation)?** The dataset has been created by the research group "Artificial Intelligence and Machine Learning" at the Computer Science Department, Technical University of Darmstadt.

**Who funded the creation of the dataset?** The dataset is created for research purposes at AIML. This work was supported by the AI lighthouse project "SPAICER" (01MK20015E), the EU ICT-48 Network of AI Research Excellence Center "TAILOR" (EU Horizon 2020, GA No 952215), and the Collaboration Lab "AI in Construction" (AICO). The work has also benefited from the Hessian Ministry of Higher Education, Research, Science and the Arts (HMWK) cluster projects "The Third Wave of AI" and "The Adaptive Mind", the Hessian Centre for Artificial Intelligence overall, the Hessian research priority program LOEWE within the project WhiteBox, and from the German Center for Artificial Intelligence (DFKI) project 'SAINT'.

### E.2 Composition

**What do the instances that comprise the dataset represent (e.g., documents, photos, people, countries)?** The dataset consists of synthetically generated images featuring simulated scenes and segmentation, depth, and metadata detailing scene composition. How many instances are there in total (of each type, if appropriate)? We have generated a total of 11 V-LoL datasets. 8 of which consists 12000 instances respectively, while the remaining 3 are used solely for out-of-distribution (OOD) testing and consist of 2,000 samples each. Does the dataset contain all possible instances or is it a sample (not necessarily random) of instances from a larger set? The datasets represent samples of an infinite set of possible arrangements. One single car sampled from the random distribution has a total of 2200 different permutations. When considering datasets consisting of trains with lengths between 2 and 4, the total number of different car samples is 23.4 trillion. As the number of cars increases, the number of possible samples grows exponentially. For a detailed description of the V-LoL generation process, please refer to Section 3.2.

**What data does each instance consist of?** Alongside the visually appealing train images, each dataset samples is annotated with detailed scene information including the individual train attributes, object masks, bounding boxes, 3D scene locations, depth information, and symbolically derived ground truth labels.

To render the V-LoL images, we utilize Python 3.10.2 and the Blender Python module version 3.3. The V-LoL 3D representation incorporates the steam locomotive ROCKET by Branislav Kubecka (blenderkit) which is under RF license (This license protects the work in the way that it allows commercial use without mentioning the author, but doesn't allow for re-sale of the asset in the same form eg. a 3D model sold as a 3D model or part of assetpack or game level on a marketplace).

**Is there a label or target associated with each instance?** The labels for each instance are symbolically derived from the underlying decision rule of the dataset.

**Is any information missing from individual instances?** No.

**Are relationships between individual instances made explicit (e.g., users' movie ratings, social network links)?** No, there are no relationships between different instances.

**Are there recommended data splits (e.g., training, development/validation, testing)?** Yes, we split 20% 80% test/train splits for the datasets, with the exception of OOD variant, which is used for evaluation only. We use stratified 5 fold cross-validation.

**Are there any errors, sources of noise, or redundancies in the dataset?** No.

**Is the dataset self-contained, or does it link to or otherwise rely on external resources (e.g.,websites, tweets, other datasets)?** The dataset is self-contained.

**Does the dataset contain data that might be considered confidential (e.g., data that is protected by legal privilege or by doctor-patient confidentiality, data that includes the content of individuals' non-public communications)?** No.

**Does the dataset contain data that, if viewed directly, might be offensive, insulting, threatening, or might otherwise cause anxiety?** No.

**Does the dataset relate to people? If not, you may skip the remaining questions in this section.** No.

**Does the dataset identify any subpopulations (e.g., by age, gender)?** NA

**Is it possible to identify individuals (i.e., one or more natural persons), either directly or indirectly (i.e., in combination with other data) from the dataset?** NA

**Does the dataset contain data that might be considered sensitive in any way (e.g., data that reveals racial or ethnic origins, sexual orientations, religious beliefs, political opinions or union memberships, or locations; financial or health data; biometric or genetic data; forms of government identification, such as social security numbers; criminal history)?** NA

### E.3 Collection Process

**How was the data associated with each instance acquired?** The data was generated.

**What mechanisms or procedures were used to collect the data (e.g., hardware apparatus or sensor, manual human curation, software program, software API)?** The images were rendered using Blender 3.3 software on generic systems and Python 3.10.2.

**If the dataset is a sample from a larger set, what was the sampling strategy (e.g., deterministic, probabilistic with specific sampling probabilities)?** See the similar question in the Composition section.

**Who was involved in the data collection process (e.g., students, crowdworkers, contractors) and how were they compensated (e.g., how much were crowdworkers paid)?** The authors were involved in the process of generating this dataset.

**Over what timeframe was the data collected?** The datasets were rendered over a period of several weeks.

**Were any ethical review processes conducted (e.g., by an institutional review board)?** No.

**Does the dataset relate to people? If not, you may skip the remainder of the questions in this section.** No.

### E.4 Preprocessing/Cleaning/Labeling

**Was any preprocessing/cleaning/labeling of the data done (e.g., discretization or bucketing, tokenization, part-of-speech tagging, SIFT feature extraction, re-**

**moval of instances, processing of missing values)?** No, the dataset was generated together with labels.

**Was the "raw" data saved in addition to the preprocessed/cleaned/labeled data (e.g., to support unanticipated future uses)?** NA

**Is the software used to preprocess/clean/label the instances available?** NA

### E.5 Uses

**Has the dataset been used for any tasks already?** In the paper we show and benchmark the intended use of this dataset for visual logical learning, specifcally evaluating on several challenges of visual logical learning.

**Is there a repository that links to any or all papers or systems that use the dataset?** No.

**What (other) tasks could the dataset be used for?** We include additional information maps when generating this dataset, which could be used for object discovery from 3D scenes. In addition the logical structure of the train representations can be converted to natural language questions for QA tasks.

**Is there anything about the composition of the dataset or the way it was collected and prepro- cessed/cleaned/labeled that might impact future uses?** No.

**Are there tasks for which the dataset should not be used?** This dataset is meant for research purposes only.

### E.6 Distribution

**Will the dataset be distributed to third parties outside of the entity (e.g., company, institution, organization) on behalf of which the dataset was created?** No.

**How will the dataset will be distributed (e.g., tarball on website, API, GitHub)?** The dataset and related evaluation code is available on the website `https://sites.google.com/view/v-lol` allowing users to download and read-in the data.

**When will the dataset be distributed?** The dataset is available now.

**Will the dataset be distributed under a copyright or other intellectual property (IP) license, and/or under applicable terms of use (ToU)?** Creative Commons CC BY 4.0 license.

**Have any third parties imposed IP-based or other restrictions on the data associated with the instances?** The V-LoL 3D representation incorporates the steam locomotive ROCKET by Branislav Kubecka (blenderkit) which is under RF license (This license protects the work in the way that it allows commercial use without mentioning the author, but doesn't allow for re-sale of the asset in the same form (eg. a 3D model sold as a 3D model or part of assetpack or game level on a marketplace). The dataset instances themselves do not have IP-based restrictions.

**Do any export controls or other regulatory restrictions apply to the dataset or to individual instances?** Not that we are are of.

### E.7 Maintenance

**Who is supporting/hosting/maintaining the dataset?** The dataset is supported by the authors and by the AIML research group. We commit to providing the necessary maintenance for the dataset to ensure the sustained integrity, quality, and accessibility of the dataset, thereby supporting continued scientific research and analysis.

**How can the owner/curator/manager of the dataset be contacted (e.g., email address)?** The authors of this dataset can be reached at their e-mail addresses: lukas_henrik.helff@tu-darmstadt.de and wolfgang.stammer@cs.tu-darmstadt.de .

**Is there an erratum?** If errors are found and erratum will be added to the website.

**Will the dataset be updated (e.g., to correct labeling errors, add new instances, delete in- stances)?** Any potential future updates or extension will be communicated via the website. The dataset will be versioned.

**If the dataset relates to people, are there applicable limits on the retention of the data associ- ated with the instances (e.g., were individuals in question told that their data would be retained for a fixed period of time and then deleted)?** NA

**Will older versions of the dataset continue to be supported/hosted/maintained?** We plan to continue hosting older versions of the dataset.

**If others want to extend/augment/build on/contribute to the dataset, is there a mechanism for them to do so?** Yes, we make the dataset generation code available.

### E.8 Other Questions

**Is your dataset free of biases?** Yes.

**Can you guarantee compliance to GDPR?** No, we are unable to comment on legal issues.

### E.9 Author Statement of Responsibility

The authors confirm all responsibility in case of violation of rights and confirm the licence associated with the dataset and its images.

