# OpenReview forum: "V-LoL: A Diagnostic Dataset for Visual Logical Learning"
_DMLR — Accepted by DMLR_

### Review · Reviewer_3pQP · 2024-01-10

**Recommendation:** 4
**Confidence:** 2

**Summary Of Contributions:**

This paper introduces a new visual logical learning dataset called V-LoL that combines visual and logical challenges. In particular, they introduce a version, V-LoL-trains, which incorporates the Michalski train problem into a visual benchmark. They use this dataset to evaluate three types of AI systems and identify the ways in which they struggle with visual logical learning problems. Both the specific subsets of the V-LoL-trains/blocks dataset used in evaluation and the infrastructure for generating more combinations are open-sourced.

**Strengths:**

Please see Strengths above. In particular, the paper introduces a significant contribution with the V-LoL trains/blocks dataset and associated generation library. The authors use this dataset to perform a high quality, extensive evaluation of performance, robustness, and generalization of three classes of models. The paper is clear and well-written, and limitations of the method are discussed clearly.

**Audience:**

Yes

**Claims And Evidence:**

Yes.

**Datasets And Benchmarks:**

The dataset is easily accessible via Github and Hugging Face with clear and extensive documentation. The data used in the paper is available for immediate download and the accompanying generation code allows the creation of more combinations of images. In addition, the License is easily identified.

There are a couple of details that are hard for me to find:
* It is unclear where a discussion of dataset maintenance is included
* In addition, I have trouble identifying what is the source / engine for the image components used in the generator. Did the authors create the visual components (train carts, wheels, backgrounds) themselves or use an external source?

**Extended Submissions:**

N/A

**Limitations:**

The Limitations are well-discussed. I have no additions beyond those listed in the Weaknesses section above.

**Requested Changes:**

This work would be strengthened by the items mentioned in the Weaknesses and Datasets and Benchmarks sections. In particular:
* Clarify / update details in the "Datasets and Benchmarks" section below, in particular dataset maintenance plan and source for visual components.
* Clarify / modify what the V-LoL contribution is, independent of the V-LoL-trains and V-LoL-blocks instantiations.
* Include additional discussion comparing findings with existing datasets.
* Include visual identifier of model types in Figures.

**Strengths And Weaknesses:**

Strengths
1. The authors introduce a new dataset that translates the Michalski train problem to the visual domain. This is a notable contribution as it allows for images with greater visual complexity than prior work while testing higher-level reasoning.
2. The dataset is designed to test models with three types of logic rules and important challenges like visual perception and logical reasoning. The authors show the impact of the new dataset in a thorough analysis of symbolic, neural, and neuro-symbolic models, which includes extensive discussion about the performance of various models.
3. V-LoL-trains includes variations in background scenes, which enables understanding of model robustness. The authors utilize this information to accompany their main evaluations with extensive analysis of generalization and test-time interventions.
4. The model evaluation in Section 3 is very thorough and includes useful insights regarding the vulnerability to label noise and data efficiency with respect to different training set sizes. The latter can also serve as a useful guideline in future uses of the benchmark.
5. In addition to the train and blocks datasets used in the paper, the authors also release code for generating new combinations of evaluation datasets, increasing the significance of their contribution.
6. The paper is clear and well-written, and well-motivated. The analysis is easy to follow. The discussion of Limitations is also useful as are the visuals/Figures employed throughout.

Weaknesses
1. The V-LoL paradigm is presented as a contribution itself (e.g. in Abstract, Introduction), with V-LoL-trains and V-LoL-blocks as the first instantiation. It'd be helpful to include more details about how the broader V-LoL paradigm is extensible to other applications beyond the train/blocks datasets, as the current presentation of V-LoL as a main contribution independent of V-LoL-trains/blocks seems a bit inflated compared to the amount of attention given to it (seemingly just the first paragraph of Section 2). This could be simply a matter of presentation that could be clarified a bit.
2. The paper would be strengthened by additional discussion more directly comparing how evaluation with the V-LoL-trains dataset allows for additional insights over existing datasets. For example, there could be a discussion of whether patterns hold for similar experiments as those in Section 3 with a subset of benchmarks show in Table 1.
3. In Figures 4, 5, 6, 7, 8, it's a bit hard to identify visually which models are members of the symbolic, neural, and neuro-symbolic groups. Some sort of visual identification would help the reader in more easily identifying patterns mentioned in the discussion of results.

---

### Review · Reviewer_dNcx · 2024-01-18

**Recommendation:** 3
**Confidence:** 2

**Summary Of Contributions:**

This paper presents the Visual Logical Learning diagnostic dataset (V-LoL), a platform designed to address the limitations of existing AI benchmarks in integrating visual perception with logical reasoning. V-LoL combines visual scenes with logical reasoning tasks, providing a testbed for a variety of AI systems, including symbolic-AI, neural-AI, and Neuro-Symbolic-AI models. The dataset's versatility is exemplified through two distinct instantiations: V-LoL-Trains and V-LoL-Blocks, both based on the Michalski train problem, but rendered in visually complex scenarios. Their contributions include the integration of logic-based inductive logic programming benchmarks with deep learning challenges, the creation of a user-friendly framework for generating large-scale visual datasets, and evaluations of AI models' capabilities in handling complex visual logical learning tasks.

**Strengths:**

I generally the development of this method makes sense. All their used components (as shown in Figure 1) are composable in a reasonable way. Overall the paper is easy to follow.

I feel not-confident to discuss "relation to prior work and relevance to the broader research community", and leave the related judgement to other reviewers and AC.

**Audience:**

Yes

**Broader Impact Concerns:**

I have no concerns on the ethical aspects for the proposed dataset.

**Claims And Evidence:**

Yes

**Datasets And Benchmarks:**

Their code are provided via https://github.com/ml-research/vlol-dataset-gen and data is listed https://huggingface.co/datasets/AIML-TUDA/v-lol-trains/tree/main/data via cc-by-4.0 license.

**Extended Submissions:**

I am unsure if this submission is a extended version or not. Can author clarify here?

**Limitations:**

Please refer to the above weakness section.

**Requested Changes:**

Please address the concerns raised in the above weakness. It generally consists of three major points:

- more Neuroal-Symbolic baselines
- The proposed benchmark may not challenge enough to examine future AI methods.
- there could be more varies about how to test generalization beyond different train numbers (training, test).

**Strengths And Weaknesses:**

- The proposed benchmark support examining various of tasks, including object recognition, counting, interpretation of spatial arrangements, comprehension of arithmetic and logical operators, and identifying and decoding intricate, chained reasoning patterns. Also, the introduction of Inductive logic programming provide a upper-bound for all tested baselines.

- Figure 3 demonstrate the proposed env supports various background scenes and rendering visualization.

- All of their experiments are provided via "depicts the average test accuracy and 95% confidence interval over 5 fold cross-validation" which could help alleviate biases. Additionally, it tested three different big kinds of AI methods.


Weaknesses:
- After scanning the experiments, my first thoughts are there are quite limited baselines. Especially, there should be more Neuro-Symbolic AI methods that deliver different type way to solve proposed benchmarks.

- I am unsure if the proposed benchmark is hard/challenge enough to examine future Neuro-Symbolic AI methods, as RCNN-Aleph shows strong performance on Visual Perception (Figure 4), Local Reasoning (Figure 5),  Test-Time Interventions (Figure 7), and Data-Efficiency (Figure 8). Only significant Label Noise (Figure 9) and generalization (Figure 6) makes this method looks worse.

- Generalization could move beyond from different train numbers to different visual environments (e.g. Figure 3. Also, how does the Neural-symbolic models pre-trained on CLEVR performs on V-LoL, vice versa? How does methods trained on V-LoL-cube transfer to V-LoL-train, and vice versa?

---

### Review · Reviewer_rqUC · 2024-04-17

**Recommendation:** 3
**Confidence:** 2

**Summary Of Contributions:**

The paper introduces V-LoL, a diagnostic dataset aimed at enhancing visual logical learning by integrating complex logical tasks with visual data, challenging current AI systems' capabilities.

**Strengths:**

See comments above

**Audience:**

Yes

**Broader Impact Concerns:**

The use of AI in contexts requiring visual and logical reasoning can have significant implications, especially in fields such as autonomous driving and medical diagnosis. This paper can discuss the ethical considerations and potential risks associated with deploying these AI models in sensitive and high-stakes environments.

**Claims And Evidence:**

(1) The claim that V-LoL can significantly improve the integration of visual perception and logical reasoning in AI systems is supported by model evaluations and comparisons across different AI methodologies.
(2) The paper demonstrated through empirical data how the dataset challenges existing AI models, yet it should better quantify the improvements in AI capabilities that can be directly attributed to the use of V-LoL.

**Datasets And Benchmarks:**

(1) V-LoL is compared with existing benchmarks, demonstrating its unique contribution to the field by filling the gap between visual complexity and logical reasoning tasks. This comparison is well-articulated and backed by comprehensive evaluations.
(2) Future versions of the paper could benefit from including comparisons with additional real-world datasets to further validate the claims of improved AI performance and robustness.

**Extended Submissions:**

N/A

**Limitations:**

(1) The synthetic nature of the visual data may not adequately represent the variability and complexity found in real-world scenarios, potentially limiting the external validity of the research findings.
(2) The evaluation of AI models primarily focuses on short-term performance metrics without considering long-term learning and adaptation strategies.

**Requested Changes:**

(1) Some sections of the paper could benefit from clearer definitions of technical terms and more detailed explanations of the logical rules used in the tests.
(2) Figures and tables throughout the paper are generally well-presented but could include more detailed captions to better guide the reader through the data.
(3) Further exploration of the impact of synthetic versus real-world imagery on model performance and generalization can be conducted to validate the dataset’s effectiveness across different environments.
(4) Some analysis of potential dataset biases and their implications for AI training can be included to ensure a holistic understanding of the model evaluations.
(5) Some explanations of logical rules and their application within the dataset can benefit the reader's understanding.

**Strengths And Weaknesses:**

Strengths:

(1) Introduced a novel dataset that integrates logical reasoning with visual data, addressing a gap in existing benchmarks.
(2) Employed a versatile framework that supports the generation of diverse and complex testing scenarios for AI models.
(3) Provided evaluation of various AI approaches, including symbolic, neural, and neuro-symbolic, offering insights into their capabilities and limitations.
(4) Made all related code and data publicly available, promoting transparency and reproducibility in research.


Weaknesses:

(1) The reliance on synthetic imagery might limit the applicability of findings to real-world scenarios where visual data are less controlled and more complex.
(2) Some AI models tested show significant limitations in generalization and handling complex logical reasoning, suggesting possible gaps in the dataset's ability to train robust models.
(3) The paper lacks a detailed discussion on the potential biases inherent in the dataset design and its impact on model training and evaluation.